# Representative Language Generation

**Charlotte Peale** [* 1]  **Vinod Raman** [* 2]  **Omer Reingold** [* 1]

## Abstract

We introduce "representative generation," extending the theoretical framework for generation proposed by Kleinberg et al. (2024) and formalized by Li et al. (2024), to additionally address diversity and bias concerns in generative models. Our notion requires outputs of a generative model to proportionally represent groups of interest from the training data. We characterize representative uniform and non-uniform generation, introducing the "group closure dimension" as a key combinatorial quantity. For representative generation in the limit, we analyze both information-theoretic and computational aspects, demonstrating feasibility for countably infinite hypothesis classes and collections of groups under certain conditions, but proving a negative result for computability using only membership queries. This contrasts with Kleinberg et al.'s (2024) positive results for standard generation in the limit. Our findings provide a rigorous foundation for developing more diverse and representative generative models.

## 1. Introduction

For decades, a central paradigm in machine learning has been prediction, where models are trained to map input data to specific output variables or categories. This approach encompasses tasks such as classification and regression, where the goal is to accurately estimate outcomes based on given inputs. However, recent years have seen a significant shift toward *generative models*, such as Large Language Models (LLMs) and diffusion-based image generators. These models are designed not to predict specific outcomes, but to create new data that resembles their training sets, offering a different approach to machine learning tasks.

This shift towards generative models necessitates the development of new theoretical frameworks to rigorously analyze their performance, capabilities, and limitations. Recently, Kleinberg & Mullainathan (2024) proposed a theoretical framework that encapsulates the fundamental objective of generative models: after being shown a sequence of strings from an unknown target language (such as all valid code snippets in java), generate new, unseen strings from the target language. Informally, we say that a model satisfies *generation in the limit* if it achieves this goal after seeing a finite number of strings from the target language. Kleinberg & Mullainathan (2024) showed that generation in the limit is indeed possible in many scenarios, such as whenever the class of potential target languages is countable, contrasting to classic negative results by Gold and Angluin on identification, which tell us that no model can identify the target language from a sequence of strings for most natural classes of languages (1967; 1979; 1980). This positive result for the task of language generation has spurred a flurry of follow-up works further formalizing the landscape of generation tasks and understanding the limits of when generation is possible (Li et al., 2024; Kalavasis et al., 2024b; Charikar & Pabbaraju, 2024; Kalavasis et al., 2024a).

While these recent results have demonstrated the possibility of successful generation, a concern remains that models that successfully generate in the limit may do so by generating from a restricted subset of the true language. For instance, consider an image generator trained on a diverse set of animal pictures. If this model were to produce only new images of cats, it would technically satisfy the criteria for generation in the limit, yet fail to capture the full diversity of the true target set of animal pictures. In the spirit of this example, it is not hard to imagine real-life concerns about the usage of LLMs and other generative models. Training data often includes text or images representing diverse groups, characterized by ethnicity, race, gender, and other protected attributes. However, the mere presence of diverse data doesn't guarantee diverse outputs. Beyond demographics, it's essential to ensure that generators intended for widespread use accurately reflect the diverse perspectives found across various communities. Ensuring proportional representation in a generator's outputs can significantly enhance its utility, enabling it to accurately reflect the diverse trends, themes, and ideological perspectives present in human discourse, rather than producing homogeneous responses.

---
[*]Equal contribution  [1]Stanford University [2]University of Michigan. Correspondence to: Charlotte Peale <cpeale@stanford.edu>, Vinod Raman <vkraman@umich.edu>.

*Proceedings of the 42^{nd} International Conference on Machine Learning*, Vancouver, Canada. PMLR 267, 2025. Copyright 2025 by the author(s).

This theoretical concern mirrors practical challenges observed in real-world generative models. One significant issue is the potential for these models to exacerbate biases present in their training data, leading to unfair representation across different demographic groups (Sheng et al., 2019; Bender et al., 2021; Kirk et al., 2021; Sheng et al., 2021; Mei et al., 2023; Gallegos et al., 2024; Zhou et al., 2024). Beyond fairness considerations, models that fail to capture the diversity of their data reduce their effectiveness and practical value. This reduced diversity is commonly observed in generative models and referred to as "mode collapse," in which a learned model focuses only on a limited subset of the true data distribution (Arjovsky et al., 2017; Goodfellow et al., 2020; Thanh-Tung & Tran, 2020; Wang et al., 2024).

In this work, we introduce a new definition of generation that protects against these concerns that we term *representative generation*. Rather than generating a single element at each step, a representative generator generates a distribution over multiple elements at each step. In addition to the now standard requirement that a generator must eventually be *consistent*, i.e. the distribution must be supported on new elements from the true language, we additionally require that all of the generator's outputted distributions be *representative*, meaning that for each group in a set of groups-of-interest, the proportion of elements from that group seen in the training sequence thus far must be close to the probability that the generator outputs a member of that group. Coming back to our animal example, if our training data stream consisted of 1/3 cats, 1/3 dogs, and 1/3 rabbits, then a representative generator with respect to groups specified by animal species could not get away with only generating cats, but would be required to generate a roughly even mix of cats, dogs, and rabbits.

We provide a detailed discussion of related notions and additional related work in Appendix A.

### 1.1. Main Contributions

The loose goal of "eventually generating consistently" has been taxonomized by prior works into a hierarchy of three notions of generation. The weakest is Kleinberg & Mullainathan (2024)'s generation in the limit (Definition 2.12), followed by the stronger notions of non-uniform generation and uniform generation introduced by Li et al. (2024) (Definitions 2.11 and 2.10, respectively).

Our results consider the feasibility of representative generation with respect to all three goals. For uniform and non-uniform generation, we focus on information-theoretic bounds, analogous to sample complexity results in learning theory, without considering computational efficiency. This approach is motivated by the recent barrier to efficiently computable non-uniform generators identified by Charikar & Pabbaraju (2024). For the weaker objective of representa-

tive generation in the limit, we analyze feasibility from both information-theoretic and computational perspectives. We prove a strong negative result, demonstrating the impossibility of achieving representative generation in the limit using only membership queries.

We give a brief overview of our main contributions below.

**Formalizing Representative Generation.** We propose a new property of generators termed *representative generation*. This property ensures that at each timestep, a generator's outputs closely approximate the proportions of training data across a collection of groups $\mathcal{A} \subseteq 2^{\mathcal{X}}$. Our definition extends the formal model of generation by Kleinberg & Mullainathan (2024), providing a rigorous framework to understand and address several real-world concerns related to generative models that we highlight below.

First, representative generation can guarantee accurate proportional representation of community opinions and perspectives found in the training data. This is crucial for maintaining the diversity of viewpoints present in the original dataset. Second, when the training data is highly diverse across the set of groups-of-interest, the generator's outputs must also exhibit high diversity. This feature protects against mode collapse and offers a tractable relaxation to existing notions of generation with breadth, which we discuss further in Appendix A.

Representative generation can also be viewed as a theoretical operationalization of some aspects of "alignment," the process of fine-tuning generative models to conform with societal and use-specific values. Imagine feeding a representative generator a gold-standard distribution of data that encapsulates a practitioner's values and diverse opinions. Unlike standard generators, which may learn to produce correct outputs but offer no guarantees about maintaining alignment with the input data distribution, representative generators are designed to both generate accurate outputs and preserve the distribution of values and perspectives found in the alignment data. This approach ensures that the model remains faithful to the intended balance of viewpoints and opinions, even as it generates novel content.

**Representative Uniform Generation.** We give a characterization of which pairs of hypothesis classes $\mathcal{H}$ and collections of groups $\mathcal{A}$ satisfy representative uniform generation, assuming the groups in $\mathcal{A}$ form a partition of $\mathcal{X}$. We characterize the representative uniform generatability of a pair $(\mathcal{H}, \mathcal{A})$ by a new combinatorial quantity that we term the *group closure dimension* (Definition 3.1):

**Theorem 1.1** (Informal Statement of Theorem 3.3). *A hypothesis class $\mathcal{H}$ and countable partition $\mathcal{A}$ can be uniformly generated with representation if and only if the group closure dimension of $(\mathcal{H}, \mathcal{A})$ is finite.*

We instantiate this result in Corollary 3.5, where we show that any finite hypothesis class $\mathcal{H}$ and finite partition $\mathcal{A}$ can satisfy representative uniform generation. This corollary can be compared with Theorem 2.2 of Kleinberg & Mullainathan (2024), who show that any finite $\mathcal{H}$ is uniformly generatable (without concern for representation).

While one might wonder if requiring representation with respect to only a finite collection of groups is always as easy as uniform generation without representation, we provide a counterexample to this conjecture in Corollary 3.6, in which we provide an example of a class $\mathcal{H}$ that is uniformly generatable, but cannot satisfy representative uniform generation with respect to a partition $\mathcal{A}$ containing just two disjoint groups.

**Representative Non-Uniform Generation.** We next consider the slightly weaker notion of representative non-uniform generation, and also give a characterization of which pairs of hypothesis classes and collections of disjoint groups can satisfy non-uniform generation with representation.

**Theorem 1.2** (Informal Statement of Theorem 3.7)**.** *A hypothesis class $\mathcal{H}$ and countable partition $\mathcal{A}$ can be non-uniformly generated with representation if and only if there exists a non-decreasing sequence of classes $\mathcal{H}_1 \subseteq \mathcal{H}_2 \subseteq \cdots$ such that $\mathcal{H} = \bigcup_{i=1}^{\infty} \mathcal{H}_i$ and $(\mathcal{H}_i, \mathcal{A})$ is uniformly generatable with representation for all $i \in \mathbb{N}$.*

We pair this characterization with a practical instantiation in Corollary 3.8, in which we show that all countable $\mathcal{H}$ and finite partitions $\mathcal{A}$ satisfy representative non-uniform generatability.

**Representative Generation in the Limit.** Finally, we consider the weakest notion of generation: representative generation in the limit. As all previous possible results also apply to this weaker setting, the results of the preceding sections already establish that any countable $\mathcal{H}$ and finite collection of disjoint groups $\mathcal{A}$ can be generated in the limit with representation. We show that the requirement that $\mathcal{A}$ be finite and disjoint can be significantly relaxed when we only require generation in the limit. In particular, in Theorem 4.4, we show that any countable collection of (possibly overlapping) groups and countable $\mathcal{H}$ satisfying an assumption we term the finite support assumption (Definition 4.2) can be generated with representation in the limit.

We show that this finite support assumption is necessary, and in Lemma 4.3 give an example of a finite $\mathcal{H}$ and countable set of disjoint groups $\mathcal{A}$ that cannot satisfy representative generation in the limit.

We finally turn to computability, and show a negative result:

**Lemma 1.3** (Informal Statement of Lemma 4.9)**.** *No generator can satisfy representative generation in the limit for all finite $\mathcal{H}$ and $\mathcal{A}$, and during each timestep use only a finite number of membership queries of the form "$x \in \mathsf{supp}(h)$?" or "$x \in A$?" for various $h \in \mathcal{H}$ and $A \in \mathcal{A}$.*

The impossibility contrasts with a positive result of Kleinberg & Mullainathan (2024), who show that generation in the limit (without representation) is possible with only a finite number of membership queries at each timestep.

## 2. Preliminaries

We consider a countable example space $\mathcal{X}$, and associate with $\mathcal{X}$ a *hypothesis class* $\mathcal{H} \subseteq \{0,1\}^{\mathcal{X}}$. Any particular $h \in \mathcal{H}$ can be thought of as an indicator for a valid element of $\mathcal{X}$ when $h(x) = 1$. In the context of languages, $h$ denotes a language with $h(x) = 1$ when $x$ is a valid string in that language.

We define the *support* of a hypothesis $h$ to be the set of all valid elements of $\mathcal{X}$ according to $h$, denoted $\mathsf{supp}(h) := \{x \in \mathcal{X} : h(x) = 1\}$. We will also sometimes reference the support of a distribution $\mu \in \Delta\mathcal{X}$, where $\mathsf{supp}(\mu) := \{x \in \mathcal{X} : \mu(x) > 0\}$ refers to the set of elements that are drawn from the distribution with positive probability. An *enumeration* of $\mathsf{supp}(h)$ is any infinite sequence $x_1, x_2, ...$ such that $\bigcup_{i \in \mathbb{N}} \{x_i\} = \mathsf{supp}(h)$.

We will make the following assumption about hypothesis classes throughout.

**Assumption 2.1** (Uniformly Unbounded Support (UUS))**.** A hypothesis class $\mathcal{H} \subseteq \{0,1\}^{\mathcal{X}}$ satisfies the *Uniformly Unbounded Support (UUS)* property if $|\mathsf{supp}(h)| = \infty$ for every $h \in \mathcal{H}$.

Going beyond the standard model of generation that has been considered in prior works, we also endow the space $\mathcal{X}$ with a countable set of (possibly overlapping) groups-of-interest $\mathcal{A} \subseteq 2^{\mathcal{X}}$. We assume that $\mathcal{X} \subseteq \bigcup_{A \in \mathcal{A}} A$, though this is without loss of generality because any $\mathcal{A}$ can be altered to satisfy this by adding a single group to the collection corresponding to the complement of the union of all existing groups.

Our results on representative uniform and non-uniform generation presented in Section 3 focus on the special case where $\mathcal{A}$ forms a partition of $\mathcal{X}$:

**Definition 2.2** (Countable Partition)**.** Let $\mathcal{X}$ be any countable example space. Then, $\mathcal{A} := \{A_1, A_2, \dots\}$ is a countable partition of $\mathcal{X}$ if and only if the following two conditions hold: (1) $A_i \cap A_j = \emptyset$ for all $i \neq j$ and (2) $\bigcup_{i=1}^{\infty} A_i = \mathcal{X}$.

We introduce two operators that will be useful for notation throughout. Note that we occasionally denote a finite sequence $x_1, ..., x_n$ by $x_{1:n}$, and use $|\{x_1, ..., x_n\}|$ to denote

the number of unique elements in $x_{1:n}$.

**Definition 2.3** (Set of Consistent Hypotheses). For any finite sequence of samples $x_{1:n}$ and hypothesis class $\mathcal{H}$, we define notation for the set of hypotheses consistent with the sample: $\mathcal{H}(x_{1:n}) := \{h \in \mathcal{H} : \{x_{1:n}\} \subseteq \mathsf{supp}(h)\}$.

**Definition 2.4** (Closure). Given a class $\mathcal{H}$ and finite sequence of samples $x_1, ..., x_n$, define the *closure operator* as the set of samples consistent with every hypothesis in $\mathcal{H}(x_1, ..., x_n)$:

$$\langle x_1, ..., x_n \rangle_{\mathcal{H}} := \begin{cases} \bigcap_{h \in \mathcal{H}(x_{1:n})} \mathsf{supp}(h), & \text{if } |\mathcal{H}(x_{1:n})| \geq 1 \\ \bot, & \text{if } |\mathcal{H}(x_{1:n})| = 0. \end{cases}$$

### 2.1. Generators

In this paper, we will consider *randomized generators*. A randomized generator is a map $\mathcal{G} : \mathcal{X}^\star \to \Delta\mathcal{X}$. Note that by definition, randomized generators output *distributions* over examples. Given a finite sequence of examples $x_1, \ldots, x_t$ and a randomized generator $\mathcal{G}$, one can always obtain a new example by sampling $\hat{x}_t \sim \mathcal{G}(x_1, \ldots, x_t)$. By allowing our generators to output distributions over examples, we can measure "representation" by comparing how close a distribution's induced group probabilities are to the empirical probabilities of the groups in the stream. To formalize this, we need a few more definitions, starting with induced group probabilities and group empirical probabilities.

**Definition 2.5** (Induced Group Probabilities). Given a distribution $\mu \in \Delta\mathcal{X}$ and countable family of groups $\mathcal{A} = \{A_i\}_{i \in \mathbb{N}}$, let $\mu|_{\mathcal{A}}$ denote $\mu$'s induced probabilities over $\mathcal{A}$ such that for any group $i \in \mathbb{N}$, we have that $\mu|_{\mathcal{A}}(i) := \Pr_{x \sim \mu}[x \in A_i]$.

Observe that for any countable partition $\mathcal{A} = \{A_i\}_{i \in \mathbb{N}}$ and $\mu \in \Delta\mathcal{X}$, $\mu|_{\mathcal{A}}$ is a probability distribution. If $\mathcal{A}$ is not a partition, the probabilities in $\mu|_{\mathcal{A}}$ may sum to more than 1.

**Definition 2.6** (Empirical Distribution and Group Empirical Probabilities). Let $x_1, \ldots, x_t$ be a finite sequence of examples, and $x_1^\star, \ldots x_m^\star$ denote its subsequence of unique examples. The *empirical distribution* of unique examples in $x_1, \ldots x_t$ is denoted by $\overline{x_{1:t}}$. In particular, for every $x \in \mathcal{X}$, we have $\overline{x_{1:t}}(x) := \frac{1}{m}\sum_{j=1}^{m} \mathbf{1}\{x_j^\star = x\}$. Given a countable collection of groups $\mathcal{A} = \{A_i\}_{i \in \mathbb{N}}$, the *group empirical probabilities* of the unique examples in $x_1, \ldots, x_t$ with respect to $\mathcal{A}$ are defined analogously as the induced group probabilities of the empirical distribution and denoted by $\overline{x_{1:t}}|_{\mathcal{A}}$.

The final ingredient is a measure of closeness between the induced group probabilities of the output of a randomized generator and the empirical group probabilities:

**Definition 2.7** (Supremum Distance). Given two vectors $\pi_1, \pi_2 \in [0, 1]^{\mathbb{N}}$ define their supremum distance as $||\pi_1 - \pi_2||_{\infty} := \max_{i \in \mathbb{N}} |\pi_1(i) - \pi_2(i)|$.

Our choice to focus on the supremum distance draws directly on the foundational principles established in algorithmic fairness literature, which emphasizes the importance of limiting the error experienced by the worst-off group. This perspective prioritizes ensuring good representation for every group rather than merely optimizing for average performance. The supremum distance naturally operationalizes this principle by measuring the maximum disparity across all groups, effectively placing an upper bound on the error that any group might experience.

It's worth noting that for finite group settings, different choices of distance measures ($L_1$ or $L_2$ distance) are typically within constant factors of one another, making the specific choice less critical. However, as we move to settings with infinite groups, these equivalences break down—distance measures such as $L_1$ become less informative, potentially obscuring significant disparities among individual groups.

While we selected the supremum distance for its robust guarantees from an algorithmic fairness perspective, exploring representation guarantees under alternative distance metrics is certainly an interesting direction for future research.

Finally, we are now able to rigorously define what it means to be $\alpha$-representative.

**Definition 2.8** ($\alpha$-Representative Generator). A randomized generator is $\alpha$-representative with respect to a countable collection of groups $\mathcal{A}$ if for every stream of examples $x_1, x_2, \ldots$ and for all $t \in \mathbb{N}$, we have that $||\mathcal{G}(x_{1:t})|_{\mathcal{A}} - \overline{x_{1:t}}|_{\mathcal{A}}||_{\infty} \leq \alpha$.

On its own, $\alpha$-representativeness is trivial to achieve – given any collection of groups $\mathcal{A}$ and any finite sequence $x_1, \ldots, x_t$, one can always output exactly the empirical distribution, $\overline{x_{1:t}}$. Thus, our goal will be to satisfy representation in addition to existing definitions of correctness for generation, as defined in Section 2.2.

### 2.2. Representative Generation

In this section, we introduce several notions of representative generatability for a tuple $(\mathcal{H}, \mathcal{A})$, each varying in the amount of distinct examples that an $\alpha$-representative generator can observe before it needs to "succeed." In our setting, a $\alpha$-representative generator "succeeds" if it is eventually "consistent." Formally, given an $\alpha$-representative generator $\mathcal{G}$, a hypothesis $h \in \mathcal{H}$, and any stream $x_1, x_2, \cdots \subseteq \mathsf{supp}(h)$, we say that $\mathcal{G}$ is *consistent* from time point $t \in \mathbb{N}$, if for all $s \geq t$:

$$\Pr_{\hat{x}_s \sim \mathcal{G}(x_{1:s})}[\hat{x}_s \in \mathsf{supp}(h) \setminus \{x_1, \ldots, x_s\}] = 1.$$

We note that generators are allowed some initial inconsistent outputs, provided they eventually achieve consistency. In

contrast, outputs must be representative of the data stream at *every* timestep. Thus, our definition implicitly prioritizes group representation over correctness in generation during initial timesteps. While consistency cannot always be verified, it is possible to construct and verify a representative distribution at each time step. This makes it reasonable to require representation throughout, and less reasonable to favor potentially consistent but verifiably unrepresentative alternatives. However, understanding the tradeoffs between representation and consistency is an interesting direction of future research.

Given our definitions of consistency and representativeness, we are now ready to introduce the strongest form of representative generatability – $\alpha$-representative uniform generatability. Roughly speaking, $\alpha$-representative uniform generatability requires that the amount of unique examples that $\mathcal{G}$ needs to observe before being consistent should be uniform over all $h \in \mathcal{H}$ and streams $x_1, x_2, \dots \subseteq \mathsf{supp}(h)$.

**Definition 2.9** ($\alpha$-Representative Uniform Generatability). Let $\mathcal{H} \subseteq \{0,1\}^{\mathcal{X}}$ be any hypothesis class satisfying the UUS property and $\mathcal{A} = \{A_i\}_{i \in \mathbb{N}}$ be any collection of groups over $\mathcal{X}$. Then, $(\mathcal{H}, \mathcal{A})$ is $\alpha$-*representatively uniformly generatable* if there exists an $\alpha$-representative generator $\mathcal{G}$ and $d^\star \in \mathbb{N}$, such that for every $h \in \mathcal{H}$ and any sequence $x_1, x_2, \dots$ with $\{x_1, x_2, \dots\} \subseteq \mathsf{supp}(h)$, if there exists $t^\star \in \mathbb{N}$ where $|\{x_1, \dots, x_{t^\star}\}| = d^\star$, then $\mathcal{G}$ is underline{consistent} from $t^\star$.

A tuple $(\mathcal{H}, \mathcal{A})$ is then representatively uniformly generatable if it is $\alpha$-representatively uniformly generatable for every $\alpha \in (0, 1]$.

**Definition 2.10** (Representative Uniform Generatability). Let $\mathcal{H} \subseteq \{0,1\}^{\mathcal{X}}$ be any hypothesis class satisfying the UUS property and $\mathcal{A} = \{A_i\}_{i \in \mathbb{N}}$ be any collection of groups over $\mathcal{X}$. Then, $(\mathcal{H}, \mathcal{A})$ is *representatively uniformly generatable*, if $(\mathcal{H}, \mathcal{A})$ is $\alpha$-representatively uniformly generatable for every $\alpha \in (0, 1]$.

Representative uniform generatability is a strong property as it requires the sample complexity (i.e. number of distinct examples before consistency is achieved) to be uniform over all choices of the adversary. We can weaken the definition by allowing the sample complexity to depend on the hypothesis, but still be uniform over all possible streams of examples the adversary might pick. This leads to our next notion of representative *non-uniform* generatability.

**Definition 2.11** (Representative Non-uniform Generatability). Let $\mathcal{H} \subseteq \{0,1\}^{\mathcal{X}}$ be any hypothesis class satisfying the UUS property and $\mathcal{A} = \{A_i\}_{i \in \mathbb{N}}$ be any countable collection of groups over $\mathcal{X}$. Then, $(\mathcal{H}, \mathcal{A})$ is *representatively non-uniformly generatable*, if for every $\alpha > 0$ there exists an $\alpha$-representative generator $\mathcal{G}$, such that for every $h \in \mathcal{H}$, there exists a $d^\star \in \mathbb{N}$ such that for any sequence $x_1, x_2, \dots$

with $\{x_1, x_2, \dots\} \subseteq \mathsf{supp}(h)$, if there exists $t^\star \in \mathbb{N}$ where $|\{x_1, \dots, x_{t^\star}\}| = d^\star$, then $\mathcal{G}$ is underline{consistent} from $t^\star$.

Finally, we can weaken the requirements even further by allowing the sample complexity to depend on both the hypothesis and stream selected by the adversary.

**Definition 2.12** (Representative Generation in the Limit). Let $\mathcal{H} \subseteq \{0,1\}^{\mathcal{X}}$ be any hypothesis class satisfying the UUS property and $\mathcal{A} = \{A_i\}_{i \in \mathbb{N}}$ be any countable collection of groups over $\mathcal{X}$. Then, $(\mathcal{H}, \mathcal{A})$ is *representatively generatable in the limit*, if for every $\alpha > 0$ there exists an $\alpha$-representative generator $\mathcal{G}$ such that for every $h \in \mathcal{H}$ and any enumeration $x_1, x_2, \dots$ of $\mathsf{supp}(h)$, there exists $t^\star \in \mathbb{N}$ such that $\mathcal{G}$ is underline{consistent} from $t^\star$.

We remark that our notions of generatability are direct analogs of those from Li et al. (2024) with an additional representation requirement.

## 3. Characterizations of Representative Uniform and Non-uniform Generation

In this section, we focus on characterizing representative uniform and non-uniform generatability. In light of the computational barriers for uniform and non-uniform generation established by Charikar & Pabbaraju (2024), our characterization is information-theoretic in nature. Moreover, we will only consider collections of groups $\mathcal{A}$ which are countable partitions of $\mathcal{X}$. This is still a very general setting, capturing problems like animal image generation, where one considers groups based on class. We leave the characterization of representative uniform and non-uniform for overlapping collections of groups as future work.

### 3.1. Representative Uniform Generation

Our starting point is representative uniform generation. In order to characterize which classes are representatively uniformly generatable, we extend the Closure dimension from Li et al. (2024) to account for the group-constraints induced by $\mathcal{A}$. To do so, we define a new scale-sensitive dimension, termed the Group Closure dimension, which accounts for the complexity induced by *both* $\mathcal{H}$ and $\mathcal{A}$.

**Definition 3.1** (Group Closure dimension). Let $\mathcal{H} \subseteq \{0,1\}^{\mathcal{X}}$ be any hypothesis class satisfying the UUS property and $\mathcal{A} = \{A_i\}_{i \in \mathbb{N}}$ be any countable partition of $\mathcal{X}$. The *Group Closure dimension* of $(\mathcal{H}, \mathcal{A})$ at scale $\alpha > 0$, denoted $\mathsf{GC}_\alpha(\mathcal{H}, \mathcal{A})$, is the largest natural number $d \in \mathbb{N}$ for which there exists *distinct* $x_1, \dots, x_d \in \mathcal{X}$ such that $\langle x_1, \dots, x_d \rangle_{\mathcal{H}} \neq \perp$ and either (1) $\max_{i \in S} \overline{x_{1:d}}|_{\mathcal{A}}(i) > \alpha$ or (2) $\alpha |\mathbb{N}/S| < \sum_{i \in S} \overline{x_{1:d}}|_{\mathcal{A}}(i)$, where $S := \{i \in \mathbb{N} : \langle x_1, \dots, x_d \rangle_{\mathcal{H}} \cap A_i \setminus \{x_1, \dots, x_d\} = \emptyset\}$. If this is true for arbitrarily large $d \in \mathbb{N}$, then we say that $\mathsf{GC}_\alpha(\mathcal{H}, \mathcal{A}) = \infty$. On the other hand, if this is not true for $d = 1$, we say that

$\mathsf{GC}_\alpha(\mathcal{H}, \mathcal{A}) = 0$.

*Remark* 3.2. Our definition of the Group Closure dimension is for countable groups $\mathcal{A} = \{A_i\}_{i \in \mathbb{N}}$. One can define the Group Closure dimension for finite groups $\mathcal{A} = \{A_i\}_{i \in [K]}$ by modifying the definition of $S$ and condition (2) in the following way. Define $S := \{i \in [K] : \langle x_1, \ldots, x_d \rangle_\mathcal{H} \cap A_i \setminus \{x_1, \ldots, x_d\} = \emptyset\}$ and then change (2) to $\alpha(K - |S|) < \sum_{i \in S} \overline{x_{1:d}}|_\mathcal{A}(i)$. We will use this modified definition when proving Corollary 3.5.

At a high-level, for any given error tolerance $\alpha > 0$, one should interpret $\mathsf{GC}_\alpha(\mathcal{H}, \mathcal{A})$ as the minimal number of distinct examples that a generator needs to see before it is guaranteed a winning strategy with respect to error level $\alpha$ (i.e. one that is both consistent and $\alpha$-representative). Indeed, as we will show in this section, $\mathsf{GC}_\alpha(\mathcal{H}, \mathcal{A})$ provides a precise quantitative bound on the sample complexity of representative uniform generation for $(\mathcal{H}, \mathcal{A})$. We highlight that unlike the Closure dimension, the Group Closure dimension depends on both $\mathcal{H}$ and $\mathcal{A}$ and is scale-sensitive and defined for every $\alpha > 0$. Scale-sensitive combinatorial dimensions are not new to learning theory, and have also been defined to characterize learnability for regression problems (Bartlett et al., 1994; Rakhlin et al., 2015).

Our first result in this section shows that for every $\alpha > 0$, the finiteness of $\mathsf{GC}_\alpha(\mathcal{H}, \mathcal{A})$ provides a qualitative characterization of $\alpha$-group constrained generatability.

**Theorem 3.3** (Characterization of $\alpha$-Representative Uniform Generatability)**.** *Let $\mathcal{X}$ be countable, $\mathcal{H} \subseteq \{0, 1\}^\mathcal{X}$ be any class satisfying the* UUS *property, and $\mathcal{A} = \{A_i\}_{i \in \mathbb{N}}$ be any countable partition of $\mathcal{X}$. Then, $(\mathcal{H}, \mathcal{A})$ is $\alpha$-representatively uniformly generatable if and only if $\mathsf{GC}_\alpha(\mathcal{H}, \mathcal{A}) < \infty$.*

An immediate consequence of Theorem 3.3 is a characterization for representative uniform generatability.

**Corollary 3.4** (Characterization of Representative Uniform Generatability)**.** *Let $\mathcal{X}$ be countable, $\mathcal{H} \subseteq \{0, 1\}^\mathcal{X}$ be any class satisfying the* UUS *property, and $\mathcal{A} = \{A_i\}_{i \in \mathbb{N}}$ be any countable partition of $\mathcal{X}$. Then, $(\mathcal{H}, \mathcal{A})$ is representatively uniformly generatable if and only if $\mathsf{GC}_\alpha(\mathcal{H}, \mathcal{A}) < \infty$ for all $\alpha > 0$.*

Our proof of Theorem 3.3 is *constructive*. In particular, to show that the finiteness of $\mathsf{GC}_\alpha(\mathcal{H}, \mathcal{A})$ is necessary, we first assume that $\mathsf{GC}_\alpha(\mathcal{H}, \mathcal{A}) = \infty$. Then, for every generator $\mathcal{G}$, we explicitly choose a hypothesis $h \in \mathcal{H}$ and construct a valid stream of examples $x_1, x_2, \cdots \subseteq \mathsf{supp}(h)$ such that $\mathcal{G}$ eventually violates either consistency or $\alpha$-representativeness. In fact, if $\mathsf{GC}_\alpha(\mathcal{H}, \mathcal{A}) = d$, a simple adaption of our proof shows that for any generator $\mathcal{G}$, there exists a hypothesis $h \in \mathcal{H}$ and a valid stream of examples $x_1, x_2, \cdots \subseteq \mathsf{supp}(h)$ such that $\mathcal{G}$ violates either consistency or $\alpha$-representativeness after observing $d$ distinct

examples. To prove that the finiteness of $\mathsf{GC}_\alpha(\mathcal{H}, \mathcal{A})$ is sufficient, we construct an $\alpha$-representative generator $\mathcal{G}$ which satisfies consistency after observing $\mathsf{GC}_\alpha(\mathcal{H}, \mathcal{A}) + 1$ distinct examples. Unlike uniform generation without representation, our generator $\mathcal{G}$ for representative uniform generation computes closures at every time point $t \in \mathbb{N}$. Combining both the necessary and sufficient directions shows that not only does the finiteness of $\mathsf{GC}_\alpha(\mathcal{H}, \mathcal{A})$ characterize $\alpha$-group constrained uniform generation, but also that the optimal sample complexity is exactly $\Theta(\mathsf{GC}_\alpha(\mathcal{H}, \mathcal{A}))$. We defer the full proof of Theorem 3.3 to Appendix C.1.

Another important consequence of Theorem 3.3 is its implications on representative uniform generation for *finite* $\mathcal{H}$ and $\mathcal{A}$. Our next result uses the Group Closure dimension and Theorem 3.3 to show that *all* tuples $(\mathcal{H}, \mathcal{A})$ where both $\mathcal{H}$ and $\mathcal{A}$ are finite are representative uniform generatable. The full proof can be found in Appendix C.2

**Corollary 3.5** (All Finite Classes and Finite Partitions are Representatively Uniformly Generatable)**.** *Let $\mathcal{X}$ be countable, $\mathcal{H} \subseteq \{0, 1\}^\mathcal{X}$ be any finite class satisfying the* UUS *property, and $\mathcal{A} = \{A_i\}_{i \in K}$ be any finite partition of $\mathcal{X}$. Then, $(\mathcal{H}, \mathcal{A})$ is representatively uniformly generatable.*

As we will show in Lemma 4.3, the finiteness of $\mathcal{A}$ is in a weak sense *necessary* for Corollary 3.5 to hold – if $\mathcal{A}$ is allowed to be countably infinite in size, there exists a hypothesis class of size one that is not even generatable in the limit with representation! Nevertheless, even when $\mathcal{A}$ is finite, representative uniform generation is still harder than (unrepresentative) uniform generation as evidenced by the following Corollary, which we prove in Appendix C.3.

**Corollary 3.6** (Representative Uniform Generation $\neq$ Uniform Generation)**.** *Let $\mathcal{X}$ be countable. There exists a countable,* UUS *class $\mathcal{H} \subseteq \{0, 1\}^\mathcal{X}$ and a finite partition $\mathcal{A} = \{A_i\}_{i \in [K]}$ of $\mathcal{X}$ such that $\mathcal{H}$ is uniformly generatable but $(\mathcal{H}, \mathcal{A})$ is* not *representatively uniformly generatable.*

In fact, Corollary 3.6 shows something stronger – there are trivially uniformly generatable classes which are *not* representatively uniformly generatable with just two groups. This brittleness of generatability when forced to satisfy both consistency and representation is not unique to our notion of diversity, but also shown by existing work on generation with breadth (Kalavasis et al., 2024b; Charikar & Pabbaraju, 2024; Kalavasis et al., 2024a).

## 3.2. Representative Non-uniform Generation

We now proceed to give a characterization of representative *non-uniform* generation. Recall that for non-uniform generation, we allow the sample complexity (i.e., the number of distinct examples needed before consistency) of the generator to depend on the hypothesis chosen by the adversary, but not the stream. Similar to the characterization of

non-uniform generatability from Li et al. (2024) without representation constraints, our main result in this section provides a characterization of representative non-uniform generation in terms of representative uniform generation.

**Theorem 3.7** (Characterization of Representative Non-uniform Generatability). *Let $\mathcal{X}$ be countable, $\mathcal{H} \subseteq \{0, 1\}^{\mathcal{X}}$ be any class satisfying the* UUS *property, and $\mathcal{A} = \{A_i\}_{i \in \mathbb{N}}$ be any countable partition of $\mathcal{X}$. Then, $(\mathcal{H}, \mathcal{A})$ is representatively non-uniformly generatable if and only if for every $\alpha > 0$, there exists a non-decreasing sequence of classes $\mathcal{H}_1 \subseteq \mathcal{H}_2 \subseteq \cdots$ such that $\mathcal{H} = \bigcup_{i=1}^{\infty} \mathcal{H}_i$ and $(\mathcal{H}_i, \mathcal{A})$ is $\alpha$-representative uniformly generatable $\forall\, i \in \mathbb{N}$.*

The proof of Theorem 3.7 is similar to the proof of Theorem 3.5 in Li et al. (2024), hence we defer the details to Appendix C.4. We instantiate Theorem 3.7 with Corollary 3.8 to show that every tuple $(\mathcal{H}, \mathcal{A})$ where $\mathcal{H}$ is countably infinite and $\mathcal{A}$ is finite is representatively non-uniformly generatable. The proof of Corollary 3.8 is in Appendix C.5. Like for representative uniform generation, the finiteness of $\mathcal{A}$ is, in a weak sense, necessary for Corollary 3.8 to hold.

**Corollary 3.8** (All Countable Classes and Finite Partitions are Representatively Non-uniformly Generatable). *Let $\mathcal{X}$ be countable, $\mathcal{H} \subseteq \{0, 1\}^{\mathcal{X}}$ be any countably infinite class satisfying the* UUS *property, and $\mathcal{A} = \{A_i\}_{i \in K}$ be any finite partition of $\mathcal{X}$. Then, $(\mathcal{H}, \mathcal{A})$ is representatively non-uniformly generatable and hence, also representatively generatable in the limit.*

# 4. Representative Generation in the Limit

We finally consider the weakest definition of successful generation: generation in the limit. As per Definition 2.12, this requires a generator to output a distribution representative of the data stream at each timestep, and after some finite point in time, all outputs must be consistent with the true language. We note that as the weakest notion of generation, all positive results from Section 3 for uniform and non-uniform representative generatability apply to representative generatability in the limit. Notably, Corollary 3.8 implies that any countably infinite $\mathcal{H}$ can be generated in the limit with representation for any finite partition $\mathcal{A}$ of $\mathcal{X}$.

In this section, we prove that relaxing to representative generatability in the limit expands the set of feasible $\mathcal{A}$'s in two ways: (1) allowing *overlapping* groups instead of disjoint partitions, and (2) permitting *countably infinite* sets of groups under a finite support assumption (defined below). We maintain that $\mathcal{X} \subseteq \bigcup_{i \in \mathbb{N}} A_i$.

**Definition 4.1** (Finite Support Size). For any hypothesis $h : \mathcal{X} \to \{0, 1\}$ and collection of possibly overlapping groups $\mathcal{A}$, we define $h$'s *finite support size* with respect to

$\mathcal{A}$ as

$$f_{h,\mathcal{A}} = \sum_{\substack{S \subseteq \mathcal{A}, \\ |\bigcap_{A \in S} A \cap \mathsf{supp}(h)| < \infty}} \left| \bigcap_{A \in S} A \cap \mathsf{supp}(h) \right|.$$

In other words, the total size of all arbitrary intersections with a subset of groups in $\mathcal{A}$ and the support of $h$. Note that in the case of disjoint groups, this quantity simplifies to the number of individuals in $\mathsf{supp}(h)$ that are members of groups that have finite intersection with $\mathsf{supp}(h)$. While any finite collection of groups $\mathcal{A}$ will always satisfy the finite support assumption, this is not the case for countably infinite sets of groups. A simple example of a collection of groups that does not satisfy the assumption is any infinite collection of groups where the size of every group is finite, such as the collection of all singletons $\mathcal{A} = \{\{x\} : x \in \mathcal{X}\}$, or the collection defined in the proof of Lemma 4.3.

**Definition 4.2** (Hypothesis Class with Finite Support). We say that a hypothesis class $\mathcal{H}$ has finite support with respect to a collection of groups $\mathcal{A}$ if for every $h \in \mathcal{H}$, $f_{h,\mathcal{A}} < \infty$.

While ideally we could show that all classes of countable groups can be generated in the limit with representation without any additional assumptions, the following lemma shows that this finite support assumption is crucially necessary for generatability with representation.

**Lemma 4.3** (Necessity of Finite Support). *There exists a countably infinite partition of $\mathcal{X}$, $\mathcal{A}$, and a finite hypothesis class $\mathcal{H}$ with $|\mathcal{H}| = 1$ that is not generatable in the limit with representation for any $0 < \alpha < 1$.*

The proof of Lemma 4.3 can be found in Appendix D.1. We contrast this impossibility result with a strong positive result: any countable $\mathcal{H}$ and countable, possibly overlapping $\mathcal{A}$ satisfying the finite support assumption can be generated in the limit with representation.

**Theorem 4.4.** *Let $\mathcal{X}$ be countable, $\mathcal{H} \subseteq \{0, 1\}^{\mathcal{X}}$ be any countable class satisfying the UUS property, and $\mathcal{A} = \{A_i\}_{i \in \mathbb{N}}$ a countable collection of possibly overlapping subsets of $\mathcal{X}$ such that $\mathcal{H}$ has finite support with respect to $\mathcal{A}$. Then, $(\mathcal{H}, \mathcal{A})$ is representative generatable in the limit.*

Before providing the proof in full, we provide a sketch of the main ideas. In Kleinberg & Mullainathan (2024), the generator they provide to generate in the limit uses the notion of a *critical hypothesis*:

**Definition 4.5** (Critical Hypothesis). Given an enumeration of $\mathcal{H}$, $h_1, h_2, ...$, we say that a hypothesis $h_n$ is *critical at step $t$* if $n \leq t$, $h_n$ is consistent with the samples seen thus far, i.e. $\{x_1, ..., x_t\} \subseteq \mathsf{supp}(h_n)$, and for every $i < n$ with $\{x_1, ..., x_t\} \subseteq \mathsf{supp}(h_i)$, we have $\mathsf{supp}(h_n) \subseteq \mathsf{supp}(h_i)$.

Given any countable $\mathcal{H}$, the generator constructed by Kleinberg & Mullainathan (2024) works as follows: at time step $t$

the generator finds the critical hypothesis $h_j$ with the largest index $j \leq t$, and outputs an arbitrary unseen element from that hypothesis's support. The correctness of their algorithm follows from a key property of the true language:

**Lemma 4.6** (Kleinberg & Mullainathan (2024), Claim 4.3)**.** *Given any countable* $\mathcal{H} = \{h_1, h_2, ...\}$ *and enumeration* $x_1, x_2, ...$ *of some* $h \in \mathcal{H}$, *there exists a timepoint* $t \in \mathbb{N}$ *such that* $h$ *is critical for all timesteps* $s \geq t$.

By the definition of a critical language, if $h$ is critical, then the critical hypothesis $h_j$ with the largest index at step $t$ must satisfy $\mathsf{supp}(h_j) \subseteq \mathsf{supp}(h)$. Thus, any point output by the generator after step $t$ must be consistent, as it comes from a subset of the true language.

While sufficient for generating in the limit, this last line of reasoning exposes an obstacle in satisfying group representation: while $h_j$ is guaranteed to be a subset of the true hypothesis's support, $\mathsf{supp}(h_j)$ could consist of only elements from a single group, even if the true hypothesis's support and the data stream thus far are more diverse. Thus, we will need to ignore some critical hypotheses that do not allow the possibility of representative generation. We define a *feasible hypothesis* to be an $h$ that admits a distribution over its unseen elements that approximates the group proportions in the empirical distribution:

**Definition 4.7** (Feasible Hypothesis)**.** Given a hypothesis $h_i \in \mathcal{H}$, we say that a $h_i$ is $\alpha$-*feasible* at step $t$ if there exists a distribution $\mu$ over unseen data points in $\mathsf{supp}(h_i) \setminus \{x_1, ..., x_t\}$ such that $\|\mu|_{\mathcal{A}} - \overline{x_{1:t}}|_{\mathcal{A}}\|_\infty \leq \alpha$.

Note that even the true hypothesis $h$ may not be $\alpha$-feasible at a given timestep. However, like criticality, we can show that there exists a timestep $d \in \mathbb{N}$ such that for all $s \geq d$, $h$ must be feasible:

**Lemma 4.8.** *Consider any countable,* UUS, $\mathcal{H} \subseteq \{0, 1\}^{\mathcal{X}}$ *and countable collection of possibly overlapping subsets* $\mathcal{A} = \{A_i\}_{i \in \mathbb{N}}$ *and assume* $\mathcal{H}$ *has finite support with respect to* $\mathcal{A}$. *Let* $x_1, x_2, ...$ *be an enumeration of* $\mathsf{supp}(h)$ *for some* $h \in \mathcal{H}$. *Then, for any* $\alpha > 0$, *there exists a* $d \in \mathbb{N}$ *such that for all* $s \geq d$, $h$ *is* $\alpha$-*feasible at timestep* $s$.

We prove this lemma in Appendix D.2. By definition, the feasibility of a language guarantees that there exists a distribution $\mu$ over unseen elements that satisfies the $\alpha$-representative requirement. Thus, we can tweak our algorithm to only consider the critical *and* feasible hypothesis $h_j$ with the largest index $j \leq t$. Combining lemmas 4.6 and 4.8, we can guarantee that for any $\alpha > 0$ there exists some finite point $t^* = \max\{d, t\}$ such that for all $s \geq t^*$, the true hypothesis $h$ is both critical and $\alpha$-feasible. Thus, for all $s \geq t^*$ at least one such feasible and critical language exists, and by the same reasoning as before, we must have $\mathsf{supp}(h_j) \subseteq \mathsf{supp}(h)$, and so outputting an $\alpha$-representative

$\mu$ from the right-most $\alpha$-feasible and critical language guarantees both consistency and representation after $t^*$. Thus, our generator satisfies representative generation in the limit.

With the building blocks of Lemmas 4.6 and 4.8 in place, the proof of Theorem 4.4 follows as described in the proof sketch. The formal proof can be found in Appendix D.3.

### 4.1. Barriers to Achieving Representative Generation in the Limit with only Membership Queries

Thus far, we have considered representative generatability only in an information-theoretic sense, without regard for the amount of computation required by our generators. However, Kleinberg & Mullainathan (2024) provide an interesting positive result: any countable $\mathcal{H}$ can be generated in the limit with a generator that only requires a finite number of membership queries of the form "$x \in \mathsf{supp}(h)$?" for any $h \in \mathcal{H}$ at each timestep. In contrast to this positive result, Charikar & Pabbaraju (2024) show that no algorithm using only membership queries can be used to non-uniformly generate for all hypothesis classes of size two.

With these results in mind, it's natural to ask whether representative generation in the limit can also be achieved by an algorithm that uses only a membership query oracle. In this new setting where we care about representation as well as consistency, it's natural to assume we can make queries about both group and hypothesis membership, i.e. ask questions of the form "$x \in \mathsf{supp}(h)$?" or "$x \in A_i$?" for any $h \in \mathcal{H}$ or $A_i \in \mathcal{A}$.

We show that unlike in the standard setting of generation in the limit, the additional representation constraint poses a significant barrier to generating with only membership queries. This negative result is with respect to the generation in the limit setting, while Charikar & Pabbaraju (2024)'s negative result holds only in the stronger non-uniform generation setting without representation. However, both proofs revolve around a similar obstacle: with only membership queries to single groups or single hypotheses, it's difficult to know whether the intersection of a group and a hypothesis's support (or in the case of Charikar & Pabbaraju (2024), the intersection of the supports of two hypotheses), contains infinitely many elements, or finitely many.

The following lemma shows that no algorithm that works even just for very simple, finite pairs of $\mathcal{H}$ and $\mathcal{A}$ can generate in the limit with representation using only membership queries. The proof proceeds by contradiction, assuming the existence of such a generator. We then analyze its behavior to construct an enumeration that forces the generator to violate either consistency or representation at each timestep. The complete proof, along with an extended discussion of special cases where representative generation with membership queries is feasible, is provided in Appendix D.4.

**Lemma 4.9** (Impossibility of Generating with Group Constraints with Only Membership Queries). *For any $\alpha < 1/2$, there cannot exist a (deterministically computed) randomized generator that satisfies $\alpha$-representative generation in the limit for any UUS hypothesis class $\mathcal{H} = \{h\}$ and finite partition of $\mathcal{X}$, $\mathcal{A} = \{A_1, A_2\}$, and uses only a finite number of membership queries of the form "$x \in \mathsf{supp}(h)$?" or "$x \in A_i$?" for $h \in \mathcal{H}$ and $i \in \{1, 2\}$ at each step.*

## 5. Discussion and Future Directions

In this paper, we introduced and analyzed the concept of *representative generatability*, which extends the theoretical framework for generation introduced by Kleinberg & Mullainathan (2024) and Li et al. (2024). This novel property ensures that the outputs of generative models closely approximate the proportions of certain groups-of-interest in the training data. We provide a complete, information-theoretic characterization of which combinations of hypothesis classes and groups are uniformly and non-uniformly generatable with representation. In addition, we study representative generatability in the limit from both information-theoretic and computational perspectives. Notably, we demonstrate that, unlike the case of non-representative generatability in the limit, membership queries alone are insufficient for computable algorithms to achieve representative generation in the limit.

Our additional constraint of representational generation highlights key tensions and possibilities between the positive results of Kleinberg & Mullainathan (2024)'s model and real-world approaches to generation. Specifically, while real-world approaches typically aim to develop generative models that closely approximate training distributions—and it is indeed natural to expect our generations to resemble training data in certain aspects–Kleinberg & Mullainathan (2024)'s notion of generation in the limit imposes no requirement that generated data must resemble previously observed data, only that it must belong to the true language. Our work maintains the generation-in-the-limit framework while introducing an additional constraint: generations must resemble training data with respect to simple statistical tests measuring the prevalence of certain subpopulations. Arguably, our notion of representation is useful to formalize even in a practical setting, as it addresses an important consideration: generative models can potentially under- or over-represent certain subpopulations, even when they demonstrate good overall alignment with the training data. There are still several directions of future work, three of which we review below.

**Representative Uniform and Non-uniform Generation for Richer Collections of Groups.** In Sections 3.1 and 3.2, we show that for all finite $\mathcal{A}$ that form a partition of $\mathcal{X}$,

all finite and countable classes are representative uniformly and non-uniformly generatable, respectively. In the case of representative generation in the limit, however, our positive results extend to a much richer class of group collections: any countable collection of possibly overlapping groups satisfying the finite support assumption. Lemma 4.3 shows that in full generality, this finite support assumption is necessary for representative generation to be possible. However, this still leaves a large gap between the collections of groups we show are uniformly and non-uniformly generatable with representation (finite partitions) vs. the collections of groups we can show are generatable in the limit with representation (countable overlapping groups with finite support). This raises the question of whether this gap can be closed: are all finite and countable classes representative uniformly and non-uniformly generatable respectively if $\mathcal{H}$ has finite support with respect to $\mathcal{A}$? If not, what are the minimal assumptions that need to be placed on $(\mathcal{H}, \mathcal{A})$ so that all finite and countable $\mathcal{H}$ are representative uniformly and non-uniformly generatable, respectively? More generally, what characterizes representative uniform and non-uniform generation when groups may be overlapping?

**Representative Generation with Dynamic Groups.** Our model of representative generation considers a fixed collection of groups $\mathcal{A}$. However, in practice, group membership may evolve with time, with existing groups growing and shrinking in size, different features signaling membership in particular groups, and even some *new* groups forming. We leave it as an important direction of future work to extend our characterizations and study of representative generation to the case where there exists a *time-indexed* collection of groups $\mathcal{A}_1, \mathcal{A}_2, \ldots$, capturing the fact that in practice, group memberships may evolve with time.

**Representative Generation Beyond the Supremum Distance.** In this paper, we quantified the quality of group representation of a generator's outputs via the supremum distance between the empirical probabilities over groups and the induced group probabilities of the generator's output. However, there are other natural ways of enforcing group representation. For example, one could consider swapping the supremum distance with the $\ell_1$-distance between group probabilities, or some other notion of distance that is more appropriate for a particular application in mind. We leave the study of generatability under these alternate notions of group representation as a important direction of future work.

## Impact Statement

This work introduces a theoretical framework for representative generation in machine learning models, addressing critical issues of bias and lack of diversity in generative model outputs. By requiring proportional representation of

different groups from training data, our approach aims to mitigate risks of underrepresentation and promote more fair and inclusive AI systems.

The implementation of representative generation could help reduce societal biases perpetuated by AI systems, potentially leading to more equitable outcomes in machine-learning-aided decision-making processes such as hiring or content recommendation. It may also enhance the utility of generative models by ensuring they capture a wider range of perspectives and experiences.

However, risks exist. Defining and categorizing groups for representation could introduce new biases or oversimplify complex social dynamics. As this work is theoretical, practical implementation will require careful consideration of these ethical implications.

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

## A. Related Works

We highlight a few existing notions that are closest to our work.

**Language Identification and Generation.**    In his seminal 1967 paper, E Mark Gold introduced the model of "Language Identification in the Limit" (1967). In this model, there is a countable collection of strings $U$ and a language family $\mathcal{L} = \{L_1, L_2, \dots\}$, where each $L_i \subseteq U$. An adversary plays a sequential game with a player over rounds indexed by $t \in \mathbb{N}$. Before the game begins, the adversary picks a language $K \in \mathcal{L}$ and an enumeration $w_1, w_2, \dots$ of the strings in $K$, with possible repetitions. In each round $t \in \mathbb{N}$, the player observes the string $w_t \in K$, and outputs an index $i_t$. The goal of the player is to eventually "identify" the language $K$ by eventually outputting indices $i_t$ such that $L_{i_t} = K$. More formally, the player has identified $K$ in the limit if there exists an $s \in \mathbb{N}$ such that $L_{i_t} = K$ for all $t \geq s$. The family $\mathcal{L}$ is identifiable in the limit, if every language $K \in \mathcal{L}$ is identifiable in the limit. Gold proved that while all finite language families $\mathcal{L}$ are identifiable in the limit, there exists simple countable language families where this is not the case. Following this work, (Angluin, 1979; 1980) provides a precise characterization of which language families are identifiable in the limit, further emphasizing the impossibility of language identification in the limit.

Very recently, Kleinberg & Mullainathan (2024) revisit the Gold's model of Language Identification in the Limit with a twist: suppose in each round $t \in \mathbb{N}$, the player is not asked to output the index of $K$, but rather a string $\hat{w} \in U$ with the hope that $\hat{w} \in K \setminus \{w_1, \dots, w_t\}$. Naming this setting "Language Generation in the Limit", Kleinberg & Mullainathan (2024) show a surprisingly different result. Unlike the case of identification, Kleinberg & Mullainathan (2024) show that all countable language families are generatable in the limit. In a follow-up work, Li et al. (2024) extend Kleinberg & Mullainathan (2024)'s results beyond language generation in the limit by (1) re-framing the problem in terms of a binary hypothesis classes defined over a countable example space (2) defining new, stronger models of generation termed uniform and non-uniform generation and (3) formalizing an abstract, prompted version of generation. In this paper, we mainly adopt the notation and models of generation from Li et al. (2024). In addition, we highlight connections between representative generation and the model of prompted generation introduced by Li et al. (2024) in Appendix B.

In addition to Li et al. (2024), there have been several follow-up works to Kleinberg & Mullainathan (2024) that study both consistency and breadth in language generation in the limit. We review the consistency results of these papers here, and defer a discussion of their results on breadth to the next section. Concurrently with Li et al. (2024), Kalavasis et al. (2024b) study generation in the stochastic setting, where the positive examples revealed to the generator are sampled i.i.d. from some unknown distribution. In this model, Kalavasis et al. (2024b) quantify the error rates for generation with consistency according to the universal rates framework of Bousquet et al. (2021). Charikar & Pabbaraju (2024) study several facets of language generation. First, they show that all countable classes satisfy the stronger property of non-uniform generation. Then, they show a hardness result – the stronger setting of non-uniform generation is not possible using only membership queries. This is in contrast to generatability in the limit, where Kleinberg & Mullainathan (2024) show that every countable class is generatable in the limit using only membership queries. Lastly, Charikar & Pabbaraju (2024) characterize which classes are uniformly generatable when additional feedback is available.

**Language Generation with Breadth.**    In their original paper, Kleinberg & Mullainathan (2024) formally describe a tension between consistency and "breadth," defined as "producing outputs that represent the full range of valid outputs in some reasonable way." This observation has prompted several follow-up studies proposing and examining various definitions of breadth. Charikar & Pabbaraju (2024) introduce the notion of "exhaustive generation," while Kalavasis et al. (2024b) and Kalavasis et al. (2024a) explore three potential definitions: "generation with breadth," "generation with approximate breadth," and "unambiguous generation." Although these notions differ slightly, they all essentially require that the generator's outputs cover nearly (or exactly) the entirety of the true language in the limit. Both sets of authors demonstrate an inherent tension between consistency and breadth, as theorized by Kleinberg & Mullainathan (2024). Notably, Kalavasis et al. (2024b) show that the strongest notion, "generation with breadth" is as difficult as identification in the limit.

Our notion of representative generation can be contrasted with these definitions of breadth as both stronger and weaker along different axes.

On one hand, representative generation can be viewed as a relaxation of generation with breadth. While it requires the generator to produce diverse outputs with respect to group membership, it doesn't demand complete coverage of the entire language. The generator has two key flexibilities: it can ignore groups that appear only in a tiny fraction of the data sequence, and it need not cover the entirety of every group it does generate from. These allowances make representative generation less

stringent than full breadth requirements. Notably, our notion proves much more tractable compared to prior breadth concepts, making it a significant relaxation. In particular, Theorem 4.4 demonstrates that any countable collection of languages and countable collection of groups, under a minor assumption, can be generated in the limit with group representation.

On the other hand, representative generation can also be seen as a stronger notion compared to breadth. Previous breadth concepts do not consider how the generated output at each step compares to the data seen thus far. For instance, a valid generator with breadth for our diverse set of animal images might produce all possible animal images in the limit but could do so by generating 1000 cat pictures for every picture of a different species. While this approach might achieve coverage in the limit, it would fail to capture the diversity of animals seen in the data stream in the short term. Such a generator would lack practical utility, as it wouldn't reflect the variety in the input data. Representative generation, in contrast, maintains diversity throughout, offering more immediate value.

**Multigroup Fairness.**   Our notion of $\alpha$-representativeness is closely related to multigroup fairness notions in the algorithmic fairness literature that have been studied mainly in the context of prediction, such as multiaccuracy and multicalibration (Hébert-Johnson et al., 2018; Kearns et al., 2018). These notions require that a predictor satisfy quality metrics like accuracy-in-expectation or calibration not only for the overall population but also for every subset within a rich collection of demographic groups. Perhaps closest to our notion is the work of Gopalan et al. (2022), who study notions of multicalibration for distributions. While they focus on the stronger notion of multicalibration for distributions, they do define an analogous notion of "$\alpha$-multiaccuracy in expectation" for distributions (Gopalan et al. (2022), Definition 9). In their framework, the requirement of an $\alpha$-representative generator can be equivalently expressed as follows: at each step, the generator must produce a distribution that is $\alpha$-multiaccurate in expectation with respect to both the empirical distribution of points generated thus far and the collection of groups $\mathcal{A}$. Our work, however, diverges from this framework by concentrating on the generative process itself. We explore how group representation can be achieved and maintained throughout the sequential generation of points, in contrast to analyzing the properties of a single estimated distribution.

**Outcome Indistinguishability.**   Although presented in terms of group representation, our notion can be reframed in the language of indistinguishability. From this perspective, the goal of representative generation can be interpreted as producing a generator that is not only consistent in the limit but also, at every step, outputs a distribution indistinguishable from the data seen so far. This indistinguishability is measured with respect to a set of tests, which in our case are precisely the indicator functions for group membership. We see potential in expanding this perspective and extending our results to incorporate richer sets of tests. This direction for future work is particularly promising given the success of related concepts in other domains. For instance, the outcome indistinguishability framework introduced by Dwork et al. (2021) for the prediction setting has proven to be a powerful notion, enabling powerful guarantees for predictors such as omniprediction (Gopalan et al., 2021).

## B. Connections to Prompted Generation

Existing works have also explored variants of the generation setting that capture generation tailored to specific prompts at each time step. This concept is referred to as "prompted generation" by Kleinberg & Mullainathan (2024), and generalized to "multiclass generation" by Li et al. (2024).

Prompted generation modifies the standard generation setting by expanding the representation of a language from a binary hypothesis $h : \mathcal{X} \to \{0, 1\}$ to a multiclass hypothesis $h : \mathcal{X} \to [k]$. At each time step, the adversary provides an element $x_t$, as well as its associated label $h(x_t)$. The generator's goal is to generate a consistent element with the same label, i.e. an $x \in \mathcal{X}$ with $h(x) = h(x_t)$.

As noted by Remark 5.1 in Li et al. (2024), the multiclass framework could be shaped to be similar to the representative generation setting by selecting a partition $\mathcal{A}$ of $\mathcal{X}$ and transforming each $h : \mathcal{X} \to \{0, 1\}$ into a multiclass hypothesis such that $h(x) = i$ if $h(x) = 1$ and $x \in A_i$, and 0 otherwise. However, the existence of such a conversion is less clear in the case of overlapping groups. Additionally, whereas representative generation only requires generations that approximate the empirical distribution, prompted generation could require the generator to output an element with a label that has appeared in only a tiny fraction of the data. For this reason, the generation guarantees for these approaches differ in nature. Prompted generation requires a certain number of elements to be seen per group before generating consistently. On the other hand, representative generation offers consistency guarantees that depend solely on the total number of elements seen, regardless of their group membership.

# C. Proofs from Section 3

## C.1. Proof of Theorem 3.3

We separate the two directions (necessity and sufficiency) of Theorem 3.3 into two proofs below.

*Proof of Necessity in Theorem 3.3.* Let $\mathcal{X}$ be countable, $\mathcal{H} \subseteq \{0,1\}^{\mathcal{X}}$ be any class satisfying the UUS property, and $\mathcal{A} = \{A_i\}_{i \in \mathbb{N}}$ be any countable infinite [1] partition of $\mathcal{X}$. Suppose that $\mathsf{GC}_\alpha(\mathcal{H}, \mathcal{A}) = \infty$. It suffices to show that for every generator $\mathcal{G}$ and $d \in \mathbb{N}$, there exists $d^\star \geq d$, an $h \in \mathcal{H}$, and a sequence $x_1, x_2, \ldots$ with $\{x_1, x_2, \ldots\} \subseteq \mathsf{supp}(h)$ such that either

(1) for every $t \in \mathbb{N}$ where $|\{x_1, \ldots, x_t\}| = d^\star$, there exists an $s \geq t$ where

$$\Pr_{\hat{x}_s \sim \mathcal{G}(x_{1:s})}[\hat{x}_s \in \mathsf{supp}(h) \setminus \{x_1, \ldots, x_s\}] < 1$$

or

(2) there exists a $t \in \mathbb{N}$ such that $||\mathcal{G}(x_{1:t})|_{\mathcal{A}} - \overline{x_{1:t}}|_{\mathcal{A}}||_\infty > \alpha$.

To that end, fix a generator $\mathcal{G}$ and a number $d \in \mathbb{N}$. Since $\mathsf{GC}_\alpha(\mathcal{H}, \mathcal{A}) = \infty$, we know there must exist some $d^\star \geq d$ and a sequence of distinct examples $x_1, \ldots, x_{d^\star}$ such that either

(a) $\exists i \in \mathbb{N}$ such that $\langle x_1, \ldots, x_{d^\star} \rangle_{\mathcal{H}} \cap A_i \setminus \{x_1, \ldots, x_{d^\star}\} = \emptyset$ and $\overline{x_{1:d^\star}}|_{\mathcal{A}}(i) > \alpha$ or

(b) $\alpha|\mathbb{N}/S| < \sum_{i \in S} \overline{x_{1:d^\star}}|_{\mathcal{A}}(i)$ where $S = \{i \in \mathbb{N} : \langle x_1, \ldots, x_{d^\star} \rangle_{\mathcal{H}} \cap A_i \setminus \{x_1, \ldots, x_{d^\star}\} = \emptyset\}$.

Consider passing to $\mathcal{G}$ the sequence $x_1, \ldots, x_{d^\star}$. If

$$\Pr_{\hat{x}_{d^\star} \sim \mathcal{G}(x_{1:d^\star})}[\hat{x}_{d^\star} \in \langle x_1, \ldots, x_{d^\star} \rangle_{\mathcal{H}} \setminus \{x_1, \ldots, x_{d^\star}\}] = 1,$$

then pick any $h \in \mathcal{H}$ such that $\{x_{1:d^\star}\} \subseteq \mathsf{supp}(h)$ and any completion of the stream $x_{d^\star+1}, x_{d^\star+2}, \ldots$ such that $\{x_{d^\star+j}\}_{j \in \mathbb{N}} \subseteq \mathsf{supp}(h) \setminus \{x_{1:d^\star}\}$. On the other hand, if

$$\Pr_{\hat{x}_{d^\star} \sim \mathcal{G}(x_{1:d^\star})}[\hat{x}_{d^\star} \in \langle x_1, \ldots, x_{d^\star} \rangle_{\mathcal{H}} \setminus \{x_1, \ldots, x_{d^\star}\}] < 1,$$

then there must exist an $h \in \mathcal{H}$ such that $\{x_{1:d^\star}\} \subseteq \mathsf{supp}(h)$ while

$$\Pr_{\hat{x}_{d^\star} \sim \mathcal{G}(x_{1:d^\star})}[\hat{x}_{d^\star} \in \mathsf{supp}(h) \setminus \{x_{1:d^\star}\}] < 1.$$

Pick this $h \in \mathcal{H}$ and complete the stream like before using this $h \in \mathcal{H}$. To see why such an $h \in \mathcal{H}$ must exist, note that if $\Pr_{\hat{x}_{d^\star} \sim \mathcal{G}(x_{1:d^\star})}[\hat{x}_{d^\star} \in \langle x_1, \ldots, x_{d^\star} \rangle_{\mathcal{H}} \setminus \{x_1, \ldots, x_{d^\star}\}] < 1$, then there must exist an $x \notin \langle x_1, \ldots, x_{d^\star} \rangle_{\mathcal{H}} \setminus \{x_1, \ldots, x_{d^\star}\}$ which $\mathcal{G}(x_{1:d^\star})$ puts positive mass on. But because this $x \notin \langle x_1, \ldots, x_{d^\star} \rangle_{\mathcal{H}} \setminus \{x_1, \ldots, x_{d^\star}\}$, there must be an $h \in \mathcal{H}$ which contains $x_{1:d^\star}$ in its support but not $x$. We now show that the selected hypothesis and stream satisfies either condition (1) or (2) in three cases.

**Case 1:** Suppose on the input $x_1, \ldots, x_{d^\star}$, we have that

$$\Pr_{\hat{x}_{d^\star} \sim \mathcal{G}(x_{1:d^\star})}[\hat{x}_{d^\star} \in \langle x_1, \ldots, x_{d^\star} \rangle_{\mathcal{H}} \setminus \{x_1, \ldots, x_{d^\star}\}] < 1.$$

Consider the hypothesis $h \in \mathcal{H}$ and the stream $x_1, x_2, \ldots$ chosen above for this case. Note that $\{x_1, x_2, \ldots\} \subseteq \mathsf{supp}(h)$ by definition. Moreover, since $x_{d^\star+1} \neq x_{d^\star}$, $t = d^\star$ is the only such time point where $|\{x_1, \ldots, x_t\}| = d^\star$. Finally, on round $s = d^\star \geq t$, we have that $\Pr_{\hat{x}_s \sim \mathcal{G}(x_{1:s})}[\hat{x}_s \in \mathsf{supp}(h) \setminus \{x_{1:s}\}] < 1$, satisfying condition (1).

---

[1] An identical proof follows if $\mathcal{A}$ is instead a finite partition of $\mathcal{X}$.

**Case 2:** Suppose on the input $x_1, \ldots, x_{d^\star}$, we have that

$$\Pr_{\hat{x}_{d^\star} \sim \mathcal{G}(x_{1:d^\star})} [\hat{x}_{d^\star} \in \langle x_1, \ldots, x_{d^\star} \rangle_{\mathcal{H}} \setminus \{x_1, \ldots, x_{d^\star}\}] = 1$$

and the input $x_1, \ldots, x_{d^\star}$ satisfies condition (a). Consider the hypothesis $h \in \mathcal{H}$ and the stream $x_1, x_2, \ldots$ chosen above for this case. Note that $\{x_1, x_2, \ldots\} \subseteq \mathrm{supp}(h)$ by definition. Let $i \in \mathbb{N}$ be the group satisfying the property in condition (a). Observe that on time point $t = d^\star$, we have that $\|\mathcal{G}(x_{1:t})|_{\mathcal{A}} - \overline{x_{1:t}}|_{\mathcal{A}}\|_\infty \geq \overline{x_{1:t}}|_{\mathcal{A}}(i) > \alpha$ because $\langle x_1, \ldots, x_{d^\star} \rangle_{\mathcal{H}} \setminus \{x_1, \ldots, x_{d^\star}\} \cap A_i = \emptyset$ and $\mathcal{G}(x_{1:d^\star})|_{\mathcal{A}}$ puts all its mass on $\langle x_1, \ldots, x_{d^\star} \rangle_{\mathcal{H}} \setminus \{x_1, \ldots, x_{d^\star}\}$. Thus, condition (2) is met and $\mathcal{G}$ violates $\alpha$-representation.

**Case 3:** Suppose on the input $x_1, \ldots, x_{d^\star}$, we have that

$$\Pr_{\hat{x}_{d^\star} \sim \mathcal{G}(x_{1:d^\star})} [\hat{x}_{d^\star} \in \langle x_1, \ldots, x_{d^\star} \rangle_{\mathcal{H}} \setminus \{x_1, \ldots, x_{d^\star}\}] = 1$$

and the input $x_1, \ldots, x_{d^\star}$ satisfies condition (b). Consider the hypothesis $h \in \mathcal{H}$ and the stream $x_1, x_2, \ldots$ chosen above for this case. Note that $\{x_1, x_2, \ldots\} \subseteq \mathrm{supp}(h)$ by construction. Let $t = d^\star$. We claim that if condition (b) holds, we have that $\|\mathcal{G}(x_{1:t})|_{\mathcal{A}} - \overline{x_{1:t}}|_{\mathcal{A}}\|_\infty > \alpha$. For the sake of contradiction, suppose that $\|\mathcal{G}(x_{1:t})|_{\mathcal{A}} - \overline{x_{1:t}}|_{\mathcal{A}}\|_\infty \leq \alpha$. Then, we have that $\mathcal{G}(x_{1:t})|_{\mathcal{A}}(i) \leq \overline{x_{1:t}}|_{\mathcal{A}}(i) + \alpha$ for all $i \in \mathbb{N}/S$ and $\mathcal{G}(x_{1:t})|_{\mathcal{A}}(i) = 0$ for all $i \in S$, where the latter is true by definition of $S$. If condition (b) holds, it must be the case that $|\mathbb{N} \setminus S| < \infty$. Thus, we can write:

$$\sum_{i \in \mathbb{N}} \mathcal{G}(x_{1:t})|_{\mathcal{A}}(i) = \sum_{i \in \mathbb{N} \setminus S} \mathcal{G}(x_{1:t})|_{\mathcal{A}}(i) \leq \sum_{i \in \mathbb{N} \setminus S} \overline{x_{1:t}}|_{\mathcal{A}}(i) + \alpha |\mathbb{N} \setminus S|.$$

If condition (b) is true, then $\sum_{i \in \mathbb{N} \setminus S} \overline{x_{1:t}}|_{\mathcal{A}}(i) < 1 - \alpha |\mathbb{N} \setminus S|$ giving that $\sum_{i \in \mathbb{N}} \mathcal{G}(x_{1:t})|_{\mathcal{A}}(i) < 1$, a contradiction to the fact that $\mathcal{G}(x_{1:t})|_{\mathcal{A}}$ is a probability measure (recall that when $\mathcal{A}$ is a partition of $\mathcal{X}$, the induced group probabilities of any $\mu \in \Delta \mathcal{X}$ form a probability measure). Thus, it must be the case that $\|\mathcal{G}(x_{1:t})|_{\mathcal{A}} - \overline{x_{1:t}}|_{\mathcal{A}}\|_\infty > \alpha$ and as in Case 2, $\mathcal{G}$ violates $\alpha$-representation and condition (2) is met.

This completes all cases. The overall proof is complete after noting that the generator $\mathcal{G}$ and number $d \in \mathbb{N}$ were picked arbitrarily. $\square$

*Proof of Sufficiency in Theorem 3.3.* Let $\mathcal{X}$ be countable, $\mathcal{H} \subseteq \{0, 1\}^{\mathcal{X}}$ be any class satisfying the UUS property, and $\mathcal{A} = \{A_i\}_{i \in \mathbb{N}}$ be any countable infinite partition of $\mathcal{X}$. Suppose that $\mathsf{GC}_\alpha(\mathcal{H}, \mathcal{A}) < \infty$ for some $\alpha > 0$. Let $d := \mathsf{GC}_\alpha(\mathcal{H}, \mathcal{A})$ Then, by definition, we have that for every $c \geq d + 1$ and sequence of distinct examples $x_1, x_2, \ldots, x_c$ such that $\langle x_1, x_2, \ldots, x_c \rangle_{\mathcal{H}} \neq \perp$, both of the following conditions hold true:

(1) $\max_{i \in S} \overline{x_{1:c}}|_{\mathcal{A}}(i) \leq \alpha$ and

(2) $\alpha |\mathbb{N}/S| \geq \sum_{i \in S} \overline{x_{1:c}}|_{\mathcal{A}}(i)$,

where $S = \{i \in \mathbb{N} : \langle x_1, \ldots, x_c \rangle_{\mathcal{H}} \cap A_i \setminus \{x_1, \ldots, x_c\} = \emptyset\}$.

We will use this fact to construct an $\alpha$-representative uniform generator satisfying the properties in Definition 2.9.

Let $x_1, x_2, \ldots$ be any stream of examples. Consider the following generator $\mathcal{G}$. For each round $t$ until $d + 1$ unique examples have been observed, $\mathcal{G}$ computes and plays from any $\mu_t \in \Delta \mathcal{X}$ such that $\mu_t|_{\mathcal{A}}$ is an $\alpha$-approximation of the group empirical distribution $\overline{x_{1:t}}|_{\mathcal{A}}$, i.e. $\|\mu_t|_{\mathcal{A}} - \overline{x_{1:t}}|_{\mathcal{A}}\|_\infty \leq \alpha$. Note that such a $\mu_t$ is always guaranteed to exist, as we can choose precisely $\mu_t = \overline{x_{1:t}}$. Suppose on round $t^\star$, we have that $|\{x_1, \ldots, x_{t^\star}\}| = d + 1$. For all rounds $t \geq t^\star$, $\mathcal{G}$ checks whether $\langle x_1, \ldots, x_t \rangle_{\mathcal{H}} = \perp$. If this is true, then $\mathcal{G}$ computes and plays from any $\mu_t \in \Delta \mathcal{X}$ such that $\mu_t|_{\mathcal{A}}$ is an $\alpha$-approximation of the group empirical distribution $\overline{x_{1:t}}|_{\mathcal{A}}$. Otherwise, $\mathcal{G}$ first computes the group empirical distribution $\overline{x_{1:t}}|_{\mathcal{A}}(i)$ and then the set $S_t = \{i \in \mathbb{N} : \langle x_1, \ldots, x_t \rangle_{\mathcal{H}} \cap A_i \setminus \{x_1, \ldots, x_t\} = \emptyset\}$. If $S_t := \emptyset$, $\mathcal{G}$ picks $z_i \in \langle x_1, \ldots, x_t \rangle_{\mathcal{H}} \cap A_i \setminus \{x_1, \ldots, x_t\}$ for all $i \in \mathbb{N}$ and constructs the distribution $\mu_t \in \Delta \mathcal{X}$ such that $\mu_t(z_i) = \overline{x_{1:t}}|_{\mathcal{A}}(i)$ and $\mu_t(x) = 0$ for all $x \in \mathcal{X} \setminus \{z_1, z_2, \ldots\}$. Otherwise, if $S_t \neq \emptyset$, $\mathcal{G}$ picks $z_i \in \langle x_1, \ldots, x_t \rangle_{\mathcal{H}} \cap A_i \setminus \{x_1, \ldots, x_t\}$ for all $i \in \mathbb{N} \setminus S_t$ and constructs a distribution $\mu_t \in \Delta \mathcal{X}$ in the following way. First, $\mathcal{G}$ picks a measure $\mu' : \mathcal{X} \to [0, 1]$ such that $\mu'_t(z_i) = \overline{x_{1:t}}|_{\mathcal{A}}(i)$ for all $i \in \mathbb{N} \setminus S_t$ and $\mu'_t(x) = 0$ for all $x \in \mathcal{X} \setminus \{z_1, z_2, \ldots\}$. At this point, $\sum_{x \in \mathcal{X}} \mu'_t(x) = \sum_{i \in \mathbb{N} \setminus S_t} \overline{x_{1:t}}|_{\mathcal{A}}(i)$ and thus, there is still $\sum_{i \in S_t} \overline{x_{1:t}}|_{\mathcal{A}}(i)$

amount of mass to be placed. To complete the distribution, $\mathcal{G}$ distributes the remaining $\sum_{i \in S_t} \overline{x_{1:t}}|_{\mathcal{A}}(i)$ mass among $\{z_1, z_2, \dots\}$ and obtains a probability measure $\mu_t \in \Delta\mathcal{X}$ such that $0 \leq \mu_t(z_i) - \overline{x_{1:t}}|_{\mathcal{A}}(i) \leq \alpha$ and $\sum_{x \in \mathcal{X}} \mu_t(x) = 1$. We will prove why this is possible below.

We now show that such a generator is $\alpha$-representative and consistent after observing $d^\star := \mathsf{GC}_\alpha(\mathcal{H}, \mathcal{A}) + 1$ distinct examples. We first prove consistency.

**Proof of consistency:** Let $\mathcal{G}$ be the generator described above. Let $h \in \mathcal{H}$ be the target hypothesis and $x_1, x_2, \dots \subseteq \mathsf{supp}(h)$ be the stream chosen by the adversary. Without loss of generality, suppose there are at least $d^\star$ distinct examples in the stream and let $t^\star$ be the first time point such that $|\{x_1, \dots, x_{t^\star}\}| = d^\star$. We need to show for all $s \geq t^\star$:

$$\Pr_{\hat{x}_s \sim \mathcal{G}(x_{1:s})} [\hat{x}_s \in \mathsf{supp}(h) \setminus \{x_1, \dots, x_s\}] = 1.$$

Fix some $s \geq t^\star$. Then, observe that by construction, $\mathcal{G}$ always picks and plays from a distribution $\mu_s \in \Delta\mathcal{X}$ such that $\mathsf{supp}(\mu_s) \subseteq \langle x_1, \dots, x_s \rangle_{\mathcal{H}} \setminus \{x_1, \dots, x_s\}$. Since $\langle x_1, \dots, x_s \rangle_{\mathcal{H}} \subseteq \mathsf{supp}(h)$, the proof of consistency is complete.

We now prove that $\mathcal{G}$ is $\alpha$-representative.

**Proof of $\alpha$-representativeness:** Fix any (not necessarily valid) sequence of examples $x_1, x_2, \dots$. It suffices to show that for every $t \in \mathbb{N}$, we have that

$$||\mathcal{G}(x_{1:t})|_{\mathcal{A}} - \overline{x_{1:t}}|_{\mathcal{A}}||_\infty \leq \alpha.$$

Let $t^\star \in \mathbb{N}$ be the smallest time point such that $|\{x_1, \dots, x_{t^\star}\}| = d^\star$. By definition, observe that $\mathcal{G}$ satisfies $\alpha$-representativeness for all $t < t^\star$. Fix some $t \geq t^\star$. There are three cases to consider. Suppose that $\langle x_1, \dots, x_t \rangle_{\mathcal{H}} = \perp$. Then by definition, $\mathcal{G}$ plays an $\alpha$-approximation of $\overline{x_{1:t}}|_{\mathcal{A}}$ and hence is $\alpha$-representative. Suppose that $\langle x_1, \dots, x_t \rangle_{\mathcal{H}} \neq \perp$ and $S_t = \emptyset$. Then, by construction, $\mathcal{G}$ picks and plays from a distribution $\mu_t \in \Delta\mathcal{X}$ such that $\mu_t|_{\mathcal{A}} = \overline{x_{1:t}}|_{\mathcal{A}}$ and thus $\alpha$-representativeness is trivially satisfied. Finally, consider the case where $\langle x_1, \dots, x_t \rangle_{\mathcal{H}} \neq \perp$ and $S_t \neq \emptyset$. We claimed above that in this scenario, $\mathcal{G}$ first computes an incomplete measure $\mu'_t$ and then distributes the remaining $\sum_{i \in S_t} \overline{x_{1:t}}|_{\mathcal{A}}(i)$ mass to obtain a probability measure $\mu_t$ such that $0 \leq \mu_t(z_i) - \overline{x_{1:t}}|_{\mathcal{A}}(i) \leq \alpha$ and $\sum_{x \in \mathcal{X}} \mu_t(x) = 1$. To see why this is possible, first note that since $|\{x_1, \dots, x_t\}| \geq \mathsf{GC}_\alpha(\mathcal{H}, \mathcal{A}) + 1$, we know that conditions (1) and (2) hold. Then, consider two cases: (i) $\sum_{i \in S_t} \overline{x_{1:t}}|_{\mathcal{A}}(i) > \alpha$ and (ii) $\sum_{i \in S_t} \overline{x_{1:t}}|_{\mathcal{A}}(i) \leq \alpha$. In case (i), we know that $\mu'_t(z_i) < 1 - \alpha$ for all $i \in \mathbb{N} \setminus S_t$. Hence, we can obtain the probability measure $\mu_t$ by adding at most $\alpha$ mass to $z_1$, and then $z_2$, and so on until all of $\sum_{i \in S_t} \overline{x_{1:t}}|_{\mathcal{A}}(i)$ has been accounted for since $\alpha |\mathbb{N}/S| \geq \sum_{i \in S_t} \overline{x_{1:t}}|_{\mathcal{A}}(i)$. In case (ii), there must exist an $i \in \mathbb{N} \setminus S_t$ such that $\mu'_t(i) \leq 1 - \sum_{i \in S_t} \overline{x_{1:t}}|_{\mathcal{A}}(i)$. Hence we can obtain the probability measure $\mu_t$, by adding all of the mass of $\sum_{i \in S_t} \overline{x_{1:t}}|_{\mathcal{A}}(i)$ on $z_i$ since $\sum_{i \in S_t} \overline{x_{1:t}}|_{\mathcal{A}}(i) \leq \alpha$. The analysis above gives that $\mathcal{G}$ plays a measure $\mu_t \in \Delta\mathcal{X}$ such that $|\mu_t|_{\mathcal{A}}(i) - \overline{x_{1:t}}|_{\mathcal{A}}(i)| \leq \alpha$ for all $i \in \mathbb{N} \setminus S_t$ and $\mu_t|_{\mathcal{A}}(i) = 0$ for all $i \in S_t$. However, since $|\{x_1, \dots, x_t\}| \geq \mathsf{GC}_\alpha(\mathcal{H}, \mathcal{A}) + 1$, condition (1) holds and thus $\max_{i \in S_t} \overline{x_{1:t}}|_{\mathcal{A}}(i) \leq \alpha$. This gives that $|\mu_t|_{\mathcal{A}}(i) - \overline{x_{1:t}}|_{\mathcal{A}}(i)| \leq \alpha$ for all $i \in S_t$ implying that $||\mathcal{G}(x_{1:t})|_{\mathcal{A}} - \overline{x_{1:t}}|_{\mathcal{A}}||_\infty \leq \alpha$. Since $t \geq t^\star$, this concludes the proof of $\alpha$-representativeness and the overall proof. $\square$

### C.2. Proof of Corollary 3.5

*Proof of Corollary 3.5.* Let $\mathcal{X}$ be countable, $\mathcal{H} \subseteq \{0,1\}^{\mathcal{X}}$ be any finite class satisfying the UUS property, and $\mathcal{A} = \{A_i\}_{i \in K}$ be any finite partition of $\mathcal{X}$. Suppose for the sake of contradiction that $\mathsf{GC}_\alpha(\mathcal{H}, \mathcal{A}) = \infty$ for some $\alpha > 0$. Let $\mathcal{F} := \{V : V \subseteq \mathcal{H}, V \neq \emptyset\}$ be the set of all non-empty subsets of $\mathcal{H}$. Since $\mathcal{H}$ is finite, $\mathcal{F}$ is also finite. We will assign a number to every $V \in \mathcal{F}$. In particular, for every $V \in \mathcal{F}$, first define and compute:

$$\langle \emptyset \rangle_V := \bigcap_{h \in V} \mathsf{supp}(h).$$

Then, let $d_V \in \mathbb{N}$ be the largest *finite* number for which there exists $x_1, \dots, x_{d_V} \in \langle \emptyset \rangle_V$ such that

(1) $\max_{i \in S} \overline{x_{1:d_V}}|_{\mathcal{A}}(i) > \alpha$ or

(2) $\alpha(K - |S|) < \sum_{i \in S} \overline{x_{1:d_V}} |_{\mathcal{A}}(i)$

where $S := \{i \in [K] : \langle \emptyset \rangle_V \setminus \{x_1, \dots, x_{d_V}\} \cap A_i = \emptyset\}$.

We will use the fact that $|K| < \infty$ to prove that such a *finite* $d_V$ exists for all $V \in \mathcal{F}$. To that end, fix some $V \in \mathcal{F}$. Without loss of generality, suppose that $|\langle \emptyset \rangle_V| = \infty$, as otherwise, the claim is trivially true (i.e if $|\langle \emptyset \rangle_V| < \infty$, it must be the case that $d_V \leq |\langle \emptyset \rangle_V| < \infty$). It suffices to show that there exists a $d \in \mathbb{N}$ such that for every $d' \geq d$ and sequence $x_1, \dots, x_{d'} \in \langle \emptyset \rangle_V$, we have that

(1) $\max_{i \in S} \overline{x_{1:d'}} |_{\mathcal{A}}(i) \leq \alpha$ and

(2) $\sum_{i \in S} \overline{x_{1:d'}} |_{\mathcal{A}}(i) \leq \alpha (K - |S|)$

where $S = \{i \in [K] : A_i \cap \langle \emptyset \rangle_V \setminus \{x_1, \dots, x_{d'}\} = \emptyset\}$. To do so, we need some more notation. First, separate the $K$ groups into two sets. The set $R_1 = \{i \in [K] : |A_i \cap \langle \emptyset \rangle_V| = \infty\}$ contains those groups whose intersection with $\langle \emptyset \rangle_V$ is unbounded in size. The set $R_2 = \{i \in [K] : |A_i \cap \langle \emptyset \rangle_V| < \infty\}$ are those groups whose intersection with $\langle \emptyset \rangle_V$ is finite. Note that for any $x_1, \dots, x_d$, the groups in $R_1$ will never appear in $S$, and so we have that $S \subseteq R_2$. Now, define

$$p := \max_{j \in R_2} |A_j \cap \langle \emptyset \rangle_V|.$$

and observe that $p < \infty$ because $|R_2| \leq K < \infty$.

We are now ready to show the existence of such a $d \in \mathbb{N}$. In particular, pick $d = \frac{Kp}{\alpha}$ and consider any sequence $x_1, \dots, x_{d'} \in \langle \emptyset \rangle_V$ for $d' \geq d$. In the worst-case, $x_1, \dots, x_{d'}$ contains $A_i \cap \langle \emptyset \rangle_V$ for all $i \in R_2$. However, for any given $i \in R_2$, at most $p$ of the elements from $x_1, \dots, x_{d'}$ can be from $A_i$. Accordingly, we have that

$$\max_{i \in S} \overline{x_{1:d'}} |_{\mathcal{A}}(i) \leq \frac{\alpha}{K} \leq \alpha.$$

Moreover, observe that

$$\sum_{i \in S} \overline{x_{1:d'}} |_{\mathcal{A}}(i) \leq \frac{\alpha |S|}{K} \leq \alpha \leq \alpha(K - |S|),$$

where the last inequality is true because the groups form a partition of $\mathcal{X}$ and so $|S| < K$. Accordingly, $d_V \leq \frac{Kp}{\alpha} < \infty$. Since $V \in \mathcal{F}$ was chosen arbitrarily, this is true for all such $V \in \mathcal{F}$. Now, we complete the proof by showing a contradiction. Define

$$d^\star = \max\{d_V : V \in \mathcal{F}\}.$$

Again, $d^\star < \infty$ because $|\mathcal{F}| < \infty$ and $d_V < \infty$ for all $V \in \mathcal{F}$. Because $\mathsf{GC}_\alpha(\mathcal{H}, \mathcal{A}) = \infty$, we know that there exists a $t \geq d^\star + 1$, and a distinct sequence of examples $x_1, \dots, x_t$ such that

(1) $\max_{i \in S} \overline{x_{1:t}} |_{\mathcal{A}}(i) > \alpha$ or

(2) $\alpha(K - |S|) < \sum_{i \in S} \overline{x_{1:t}} |_{\mathcal{A}}(i)$

where $S := \{i \in \mathbb{N} : \langle x_1, \dots, x_t \rangle_{\mathcal{H}} \cap A_i \setminus \{x_1, \dots, x_t\} = \emptyset\}$. Take $V^\star := \mathcal{H}(x_1, \dots, x_t)$ and note that $\langle x_1, \dots, x_t \rangle_{\mathcal{H}} = \langle \emptyset \rangle_{V^\star}$. Thus, we have shown that $d_{V^\star} \geq t \geq d^\star + 1$, which contradicts the fact that $d^\star = \max\{d_V : V \in \mathcal{F}\}$ since $V^\star \in \mathcal{F}$. The proof is complete, showing that $\mathsf{GC}_\alpha(\mathcal{H}, \mathcal{A}) < \infty$. $\qquad \square$

### C.3. Proof of Corollary 3.6

*Proof of Corollary 3.6.* Let $\mathcal{X} = \mathbb{Z}$ be the set of integers. Let $\mathcal{H}' \subseteq \{0,1\}^{\mathbb{N}}$ be any class that is not uniformly generatable (for example see Lemma 3.9 in Li et al. (2024)). Define a new class $\mathcal{H} \subseteq \{0,1\}^{\mathcal{X}}$ such that $\mathcal{H} := \{x \mapsto \mathbf{1}\{x \in \operatorname{supp}(h') \cup \mathbb{Z}_{\leq 0}\} : h' \in \mathcal{H}'\}$ and let $\mathcal{A} := \{\mathbb{N}, \mathbb{Z}_{\leq 0}\}$. Note that $\mathcal{A}$ is finite partition of $\mathcal{X}$ of size 2.

First, observe that $\mathcal{H}$ is trivially uniformly generatable since $\left|\bigcap_{h \in \mathcal{H}} \operatorname{supp}(h)\right| \geq \left|\mathbb{Z}_{\leq 0}\right| = \infty$. We now show that $(\mathcal{H}, \mathcal{A})$ is not representatively uniformly generatable. For the sake of contradiction, suppose that $(\mathcal{H}, \mathcal{A})$ was representatively uniformly generatable. Then, for every $\alpha \in (0,1]$, there exists an $\alpha$-representative generator $\mathcal{G}$ and a number $d^{\star} \in \mathbb{N}$ such that after observing $d^{\star}$ distinct examples, the output of $\mathcal{G}$ is consistent (see Definition 2.9). Let $\mathcal{G}$ be such a generator for any $\alpha < 1$. We will use $\mathcal{G}$ in a blackbox manner to construct a uniform generator for $\mathcal{H}'$.

Consider the following generator $\mathcal{G}'$ for $\mathcal{H}'$. On input $x_1, \dots, x_t \in \mathbb{N}$, $\mathcal{G}'$ passes $x_1, \dots, x_t$ to $\mathcal{G}$ and receives an $\alpha$-representative distribution $\mu_t$. $\mathcal{G}'$ plays any $x \in \operatorname{supp}(\mu_t) \cap \mathbb{N} \setminus \{x_1, \dots, x_t\}$ if one exists. Otherwise, $\mathcal{G}'$ plays any $x \in \mathcal{X}$. We claim that $\mathcal{G}'$ perfectly generates from $\mathcal{H}'$ after observing $d^{\star}$ number of distinct examples. To see why, let $h' \in \mathcal{H}'$ and $x_1, \dots, x_t \subset \operatorname{supp}(h')$ be any sequence such that $|\{x_1, \dots, x_t\}| \geq d^{\star}$. Let $h : \mathcal{X} \to \{0,1\}$ be defined as $h(x) := \mathbf{1}\{x \in \operatorname{supp}(h') \cup \mathbb{Z}_{\leq 0}\}$. Then, $h \in \mathcal{H}$, $\{x_1, \dots, x_t\} \subseteq \operatorname{supp}(h)$, and therefore by definition of $d^{\star}$ and $\mathcal{G}$, we have that

(1) $\Pr_{\hat{x}_t \sim \mu_t}[\hat{x}_t \in \operatorname{supp}(h) \setminus \{x_1, \dots, x_t\}] = 1$ and

(2) $||\mu_t|_{\mathcal{A}} - \overline{x_{1:t}}|_{\mathcal{A}}||_{\infty} \leq \alpha$.

Because $x_{1:t} \subset \mathbb{N}$ and $\alpha < 1$, in order to satisfy conditions (1) and (2), there must exist an $\hat{x}_t \in \operatorname{supp}(\mu_t) \cap \mathbb{N} \setminus \{x_1, \dots, x_t\}$, and hence $\mathcal{G}'$, by playing this $\hat{x}_t$, perfectly generates on round $t$. Since $t \in \mathbb{N}$ and $x_1, \dots, x_t$ were chosen arbitrarily, we have that $\mathcal{G}'$ is a uniform generator for $\mathcal{H}'$ which contradicts the fact that $\mathcal{H}'$ was chosen to not be uniformly generatable. □

### C.4. Proof of Theorem 3.7

We separate the two directions (necessity and sufficiency) of Theorem 3.7 into two proofs below.

*Proof of Sufficiency in Theorem 3.7.* Let $\mathcal{X}$ be countable, $\mathcal{H} \subseteq \{0,1\}^{\mathcal{X}}$ be any class satisfying the UUS property, and $\mathcal{A} = \{A_i\}_{i \in \mathbb{N}}$ be any countable partition of $\mathcal{X}$.

Fix some $\alpha > 0$. Suppose there exists a non-decreasing sequence of classes $\mathcal{H}_1 \subseteq \mathcal{H}_2 \subseteq \cdots$ such that $\mathcal{H} = \bigcup_{i=1}^{\infty} \mathcal{H}_i$ and $(\mathcal{H}_i, \mathcal{A})$ is $\alpha$-representative uniformly generatable for all $i \in \mathbb{N}$. Then, there exists a $\alpha$-representative uniform generator $\mathcal{G}_i$ for each $(\mathcal{H}_i, \mathcal{A})$. For every $i \in \mathbb{N}$, let $n_i$ be the number of distinct examples that $\mathcal{G}_i$ needs to see before it is consistent (i.e. $n_i$ is the $d^{\star}$ in Definition 2.9). Observe we can assume without loss of generality that $n_1, n_2, \dots$ is a non-decreasing sequence by using the $\alpha$-representative uniform generator constructed in the proof of Theorem 3.3.

We will now use $\mathcal{G}_1, \mathcal{G}_2, \dots$ to construct a $\alpha$-representative non-uniform generator $\mathcal{Q}$. To that end, let $x_1, x_2, \dots$ be any stream of examples. Consider the following generator $\mathcal{Q}$. On time point $t \in \mathbb{N}$, $\mathcal{Q}$ first computes the number of distinct examples $d_t := |\{x_1, \dots, x_t\}|$ in the stream so far. Then, $\mathcal{Q}$ computes

$$i_t = \max \{i \in [t] : n_i \leq d_t\} \cup \{1\}$$

and plays $\mathcal{G}_{i_t}(x_1, \dots, x_t)$, the output of $\mathcal{G}_{i_t}$ on input $x_1, \dots, x_t$.

We now show that $\mathcal{Q}$ is an $\alpha$-representative non-uniform generator that satisfies Definition 2.11. Let us start by proving that $\mathcal{Q}$ is $\alpha$-representative. Let $x_1, x_2, \dots$ be any (not necessarily valid) stream of examples. It suffices to show that for every $t \in \mathbb{N}$, we have that $||\mathcal{Q}(x_{1:t})|_{\mathcal{A}} - \overline{x_{1:t}}|_{\mathcal{A}}||_{\infty} \leq \alpha$. Recall that $\mathcal{Q}(x_1, \dots, x_t) = \mathcal{G}_{i_t}(x_1, \dots, x_t)$. Since $\mathcal{G}_{i_t}$ is an $\alpha$-representative generator for $(\mathcal{H}_{i_t}, \mathcal{A})$, it must be the case that

$$||\mathcal{G}_{i_t}(x_{1:t})|_{\mathcal{A}} - \overline{x_{1:t}}|_{\mathcal{A}}||_{\infty} \leq \alpha.$$

We now complete the proof by establishing the consistency guarantees of $\mathcal{Q}$. To that end, let $h \in \mathcal{H}$ and $x_1, x_2, \dots \subseteq \operatorname{supp}(h)$ be the hypothesis and stream picked by the adversary. Let $j^{\star} \in \mathbb{N}$ be the smallest index such that $h \in \mathcal{H}_{j^{\star}}$ and let

$d^\star := \max\{j^\star, n_{j^\star}\}$. We claim that $\mathcal{Q}$ is consistent after observing $d^\star$ unique examples. To see why, suppose without loss of generality that $x_1, x_2, \ldots$ contains at least $d^\star$ distinct elements and let $t^\star$ be the smallest time point such that $|\{x_1, \ldots, x_{t^\star}\}| = d^\star$. We need to show that for all $s \geq t^\star$, we have that $\Pr_{\hat{x}_s \sim \mathcal{Q}(x_{1:s})}[\hat{x}_s \in \mathsf{supp}(h) \setminus \{x_1, \ldots, x_s\}] = 1$. Fix some $s \geq t^\star$. Then, observe that $d_s \geq n_{j^\star}$ and $t^\star \geq j^\star$. Accordingly, we have that $i_s \geq j^\star$ and hence $h \in \mathcal{H}_{i_s}$. Since $n_{i_s} \leq d_s$, we also know that

$$\Pr_{\hat{x}_s \sim \mathcal{G}_{i_s}(x_{1:s})}[\hat{x}_s \in \mathsf{supp}(h) \setminus \{x_1, \ldots, x_s\}] = 1.$$

Finally, since $\mathcal{Q}(x_{1:s}) = \mathcal{G}_{i_s}(x_{1:s})$ by construction, we have that $\mathcal{Q}$ is consistent on round $s$. Since $s \geq t^\star$ is chosen arbitrarily, this is true for all such $s$, completing the proof of consistency and the overall. proof that $\mathcal{Q}$ is an $\alpha$-representative non-uniform generator for $(\mathcal{H}, \mathcal{A})$. $\square$

*Proof of Necessity in Theorem 3.7.* Let $\mathcal{X}$ be countable, $\mathcal{H} \subseteq \{0,1\}^{\mathcal{X}}$ be any class satisfying the UUS property, and $\mathcal{A} = \{A_i\}_{i \in \mathbb{N}}$ be any countable partition of $\mathcal{X}$. Suppose that $(\mathcal{H}, \mathcal{A})$ is representative non-uniformly generatable. We need to show that for every $\alpha > 0$, there exists a non-decreasing sequence of classes $\mathcal{H}_1 \subseteq \mathcal{H}_2 \subseteq \cdots$ such that $\mathcal{H} = \bigcup_{i=1}^{\infty} \mathcal{H}_i$ and $(\mathcal{H}_i, \mathcal{A})$ is $\alpha$-representative uniformly generatable $\forall i \in \mathbb{N}$. To that end, fix some $\alpha > 0$. If $\mathcal{H}$ is representative non-uniformly generatable, then for this error level $\alpha$, there exists a $\alpha$-representative non-uniform generator $\mathcal{G}$. For every $h \in \mathcal{H}$, let $d_h \in \mathbb{N}$ be such that for any sequence $x_1, x_2, \ldots$ with $\{x_1, x_2, \ldots\} \subseteq \mathsf{supp}(h)$, if there exists $t^\star \in \mathbb{N}$ where $|\{x_1, \ldots, x_{t^\star}\}| = d_h$, then $\mathcal{G}$ is consistent from $t^\star$. Now, define $\mathcal{H}_i := \{h \in \mathcal{H} : d_h \leq i\}$ for all $i \in \mathbb{N}$. Note that $\mathcal{H}_1 \subseteq \mathcal{H}_2 \subseteq \cdots$ and that $\bigcup_{i=1}^{\infty} \mathcal{H}_i = \mathcal{H}$. Finally, observe that $\mathcal{G}$ is an $\alpha$-representative uniform generator for $\mathcal{H}_i$, and hence, $(\mathcal{H}_i, \mathcal{A})$ is $\alpha$-representative uniformly generatable. $\square$

## C.5. Proof of Corollary 3.8

*Proof.* Let $\mathcal{X}$ be countable, $\mathcal{H} \subseteq \{0,1\}^{\mathcal{X}}$ be any countably infinite class satisfying the UUS property, and $\mathcal{A} = \{A_i\}_{i \in K}$ be any finite partition of $\mathcal{X}$. Fix some $\alpha > 0$. By Theorem 3.7, it suffices to show that there exists a non-decreasing sequence of classes $\mathcal{H}_1 \subseteq \mathcal{H}_2 \subseteq \cdots$ such that $\mathcal{H} = \bigcup_{i=1}^{\infty} \mathcal{H}_i$ and $(\mathcal{H}_i, \mathcal{A})$ is $\alpha$-representative uniformly generatable $\forall i \in \mathbb{N}$. Let $h_1, h_2, \ldots$ be any enumeration of $\mathcal{H}$. Define $\mathcal{H}_i = \{h_1, \ldots, h_i\}$ for all $i \in \mathbb{N}$. Observe that $\mathcal{H}_1 \subseteq \mathcal{H}_2 \subseteq \cdots$ and that $\mathcal{H} = \bigcup_{i=1}^{\infty} \mathcal{H}_i$. Finally, since $|\mathcal{H}_i| = i < \infty$ and $|\mathcal{A}| < \infty$, by Corollary 3.5, we have that $(\mathcal{H}_i, \mathcal{A})$ is $\alpha$-representative uniformly generatable. Since representative non-uniform generatability implies representative generatability in the limit, our proof is complete. $\square$

# D. Proofs from Section 4

## D.1. Proof of Lemma 4.3

*Proof of Lemma 4.3.* Select any $0 < \alpha < 1$, and let $\mathcal{H} = \{h\}$, where $\mathsf{supp}(h) = \mathcal{X}$ (note that this $\mathcal{H}$ is trivially generatable in the limit if we do not require group representation). Consider any arbitrary enumeration of $\mathsf{supp}(h)$, $u_1, u_2, \ldots$, with each $u_i$ distinct. Define a countable partition of $\mathcal{X}$, $\mathcal{A} = \{A_1, A_2, \ldots\}$ such that for each $i \in \mathbb{N}$,

$$A_i = \left\{ u_j : \sum_{k=0}^{i-1} \left(\frac{1}{1-\alpha}\right)^k \leq j < \sum_{k=0}^{i} \left(\frac{1}{1-\alpha}\right)^k \right\}.$$

Clearly this is a valid partition as all groups are disjoint, and because $\frac{1}{1-\alpha} > 1$ by definition, for every $u_j$, there exists an $i \in \mathbb{N}$ such that $\sum_{k=0}^{i-1} \left(\frac{1}{1-\alpha}\right)^k \leq j < \sum_{k=0}^{i} \left(\frac{1}{1-\alpha}\right)^k$. Note that for each $i \in \mathbb{N}$,

$$|A_i| = |A_i \cap \mathsf{supp}(h)| = \sum_{k=0}^{i} \left(\frac{1}{1-\alpha}\right)^k - \sum_{k=0}^{i-1} \left(\frac{1}{1-\alpha}\right)^k = \left(\frac{1}{1-\alpha}\right)^i.$$

Choose $u_1, u_2, \ldots$ be the enumeration of $\mathsf{supp}(h)$ selected by the adversary. Consider any $i \in \mathbb{N}$, and note that at step $t = \sum_{j=0}^{i} \left(\frac{1}{1-\alpha}\right)^j - 1$, all elements of $A_i$ have been exhausted, and $\mathsf{supp}(h) \cap A_i \setminus \{x_1, \ldots, x_t\} = \emptyset$. Moreover, $A_i$ takes

up more than an $\alpha$-fraction of the distinct elements seen thus far, as $\overline{x_{1:t}}|_{\mathcal{A}}(i)$ can be lower-bounded as

$$
\begin{aligned}
\frac{|A_i \cap x_{1:t}|}{|x_{1:t}|} &= \frac{\left(\frac{1}{1-\alpha}\right)^i}{\sum_{j=0}^i \left(\frac{1}{1-\alpha}\right)^j - 1} \\
&= \frac{\left(\frac{1}{1-\alpha}\right)^i}{\frac{\frac{1}{1-\alpha}(1-(\frac{1}{1-\alpha})^i)}{1-\frac{1}{1-\alpha}}} \\
&= \frac{\left(\frac{1}{1-\alpha}\right)^{i+1} - \left(\frac{1}{1-\alpha}\right)^i}{\left(\frac{1}{1-\alpha}\right)^{i+1} - \frac{1}{1-\alpha}} \\
&> \frac{\left(\frac{1}{1-\alpha}\right) - 1}{\left(\frac{1}{1-\alpha}\right)} \\
&= \alpha
\end{aligned}
$$

formula for sum of geometric series

This implies that in order to be representative, the generator must output a distribution $\mu_t$ that puts positive mass on some $x \in A_i$, otherwise

$$
|\mu_t|_{\mathcal{A}}(i) - \overline{x_{1:t}}|_{\mathcal{A}}(i)| \geq \overline{x_{1:t}}|_{\mathcal{A}}(i) > \alpha.
$$

However, because all elements of $A_i$ have already been exhausted in the sequence, the generator must either violate consistency by putting mass on an $x \in A_i$ that has already been seen in the sequence, or violate representation by putting no mass on $A_i$ despite appearing in greater than an $\alpha$-fraction of the sequence. Thus, the generator cannot satisfy both consistency and representation simultaneously at this timestep (in fact, it cannot generate correctly from *any* of the groups that have been seen so far).

This happens for each $A_i$, and thus any generator must make infinitely many mistakes at timesteps $\sum_{j=0}^i \left(\frac{1}{1-\alpha}\right)^j - 1$ for every $i \in \mathbb{N}$ on this sequence, and thus it cannot satisfy the requirement of representative generation in the limit. $\qquad\square$

### D.2. Proof of Lemma 4.8

*Proof of Lemma 4.8.* We introduce the notion of a group vector $v(x) \in \{0,1\}^{\mathbb{N}} \setminus \{0^{\mathbb{N}}\}$, which given $x \in \mathcal{X}$, indicates an $x$'s group membership in all groups in $\mathcal{A}$, with $v(x)_i = \mathbf{1}[x \in A_i]$.

Recall that $f_{h,\mathcal{A}}$ denotes the finite support size of $h$ with respect to $\mathcal{A}$ (Definition 4.1), which by assumption satisfies $f_{h,\mathcal{A}} < \infty$. We can equivalently write the definition of $f_{h,\mathcal{A}}$ in terms of group membership vectors, i.e.

$$
f_{h,\mathcal{A}} = \sum_{\substack{v \in \{0,1\}^{\mathbb{N}} \setminus \{0^{\mathbb{N}}\}, \\ |\bigcap_{i \in \mathbb{N}, v_i=1} A_i \cap \mathsf{supp}(h)| < \infty}} \left| \bigcap_{i \in \mathbb{N}, v_i=1} A_i \cap \mathsf{supp}(h) \right|.
$$

Let $V \subseteq \{0,1\}^{\mathbb{N}} \setminus \{0^{\mathbb{N}}\}$ be the subset of all group membership vectors with finite support, i.e. all $v \in \{0,1\}^{\mathbb{N}} \setminus \{0^{\mathbb{N}}\}$ such that

$$
\left| \bigcap_{i \in \mathbb{N}, v_i=1} A_i \cap \mathsf{supp}(h) \right| < \infty.
$$

The remainder of the proof follows in three key steps:

1. We first show that after enough timesteps, the proportion of the data sequence taken up by unique elements with group membership lying in $V$ must shrink to a very small quantity and stay below that amount. In particular, less than $\alpha/2$.

2. Next, we show that due to how little mass is placed on these vectors, it's possible to construct a distribution that $\alpha$-approximates the empirical group probabilities, but places no mass on any $x$ with $v(x) \in V$.

3. It follows that $h$ must be feasible at this point, because all other group vectors have infinite support when intersected with $\text{supp}(h)$.

**Step 1: $V$ takes up a small proportion of of the empirical distribution.** Consider the timestep $t$ at which point the sequence has included $d^* = \frac{2 f_{h,\mathcal{A}}}{\alpha}$ unique elements. For any $s \geq t$, the proportion of elements in the empirical distribution with $v(x) \in V$ is upper bounded by

$$\frac{f_{h,\mathcal{A}}}{d^*} = \alpha/2.$$

Thus, for all $s \geq t$, the total proportion of unique elements from $V$ in the sequence must be at most $\alpha/2$, as desired.

**Step 2: Constructing an $\alpha$-approximate distribution that ignores $V$.** We now show that for all $s \geq t$, because we are guaranteed that the total proportion of unique datapoints from $V$ is at most $\alpha/2$, we can construct a distribution whose group probabilities $\alpha$-approximate $\overline{x_{1:s}}|_{\mathcal{A}}$ and places no mass on any $x$ with $v(x) \in V$. For now we will ignore exactly what elements the distribution uses, and just care about their group membership vectors. We construct a distribution $\pi_s$ over these elements as follows:

$$\pi_s(x) \propto \begin{cases} 0 & v(x) \in V \text{ or } x \notin x_{1:s} \\ \overline{x_{1:s}}(x) & \text{otherwise} \end{cases}.$$

Clearly by definition $\pi_s$ places no mass on any $x$ with $v(x) \in V$. It remains to show that this is actually a valid approximation, i.e $\|\pi_s|_{\mathcal{A}} - \overline{x_{1:s}}|_{\mathcal{A}}\|_\infty \leq \alpha$.

To this end, consider any $A_i \in \mathcal{A}$. We rewrite the proportions of $A_i$ appearing in $\pi_s$ and the empirical distribution:

$$
\begin{aligned}
&|\pi_s|_{\mathcal{A}}(i) - \overline{x_{1:s}}|_{\mathcal{A}}(i)| \\
&\leq \Pr_{x \sim \overline{x_{1:s}}}[x \in A_i, v(x) \in V] + \frac{\Pr_{x \sim \overline{x_{1:s}}}[v(x) \in V]}{1 - \Pr_{x \sim \overline{x_{1:s}}}[v(x) \in V]} \Pr_{x \sim \overline{x_{1:s}}}[x \in A_i, v(x) \notin V] \\
&\leq \Pr_{x \sim \overline{x_{1:s}}}[x \in A_i, v(x) \in V] + \frac{\Pr_{x \sim \overline{x_{1:s}}}[v(x) \in V]}{1 - \Pr_{x \sim \overline{x_{1:s}}}[v(x) \in V]} \Pr_{x \sim \overline{x_{1:s}}}[v(x) \notin V] \\
&\leq 2 \Pr_{x \sim \overline{x_{1:s}}}[v(x) \in V] \\
&\leq \alpha \qquad\qquad\qquad\qquad\qquad\qquad\qquad\qquad\qquad\qquad\qquad\qquad\qquad \text{guarantee from Step 1}
\end{aligned}
$$

Thus, because this holds for any $A_i$, we conclude that $\pi_s|_{\mathcal{A}}$ $\alpha$-approximates $\overline{x_{1:s}}|_{\mathcal{A}}$.

**Step 3: Constructing a feasible distribution.** $\pi_s$ passes all group representation requirements, but is not quite what we want, because $\text{supp}(\pi_s)$ is supported on previously seen points in $x_{1:s}$, whereas we want a distribution supported on $\text{supp}(h) \setminus x_{1:s}$. Note that it suffices if for every $x \in \text{supp}(\pi_s)$ we can find an $x' \in \text{supp}(h) \setminus x_{1:s}$ such that $v(x) = v(x')$. Then, the distribution $\mu_s$ defined as $\mu_s(x') = \pi_s(x)$ would exactly match the group vector proportions of $\pi_s$, and thus also satisfy $\|\mu_s|_{\mathcal{A}} - \overline{x_{1:s}}|_{\mathcal{A}}\|_\infty \leq \alpha$ as well as $\text{supp}(\mu_s) \subseteq \text{supp}(h) \setminus x_{1:s}$.

This is in fact easy to do, as note that by construction, for every $x \in \text{supp}(\pi_s)$, we must have $v(x) \notin V$, and thus

$$\left| \bigcap_{i \in \mathbb{N}, v(x)_i = 1} A_i \cap \text{supp}(h) \right| = \infty.$$

This means that even after removing $x_{1:s}$, there are an infinite number of $x'$ that we can choose from to replace $x$ for each $x \in \text{supp}(\pi_s)$.

Thus, putting all these steps together, we conclude that at the finite timestep $t \in \mathbb{N}$ after we have seen $2 f_{h,\mathcal{A}}/\alpha$ unique elements, for every $s \geq t$ we can find a distribution $\mu_s$ with $\text{supp}(\mu_s) \subseteq \text{supp}(h) \setminus x_{1:s}$ and $\|\mu_s|_{\mathcal{A}} - \overline{x_{1:s}}|_{\mathcal{A}}\|_\infty \leq \alpha$. Thus, $h$ is $\alpha$-feasible for all $s \geq t$, completing the proof. $\qquad \square$

### D.3. Proof of Theorem 4.4

*Proof of Theorem 4.4.* Choose some $\alpha > 0$, and consider the following mapping from a sequence of examples $x_1, ..., x_t$ to a distribution over $\mathcal{X}$:

1. Given examples $x_1, ..., x_t$, let $C_t \subseteq \{h_1, ..., h_t\}$ be the set of critical hypotheses at step $t$.

2. Let $F_t \subseteq C_t$ be the subset of critical hypotheses that are also $\alpha$-feasible.

3. if $F_t$ is empty, output the distribution $\mu_t = \overline{x_{1:t}}$.

4. Otherwise, let $h_n \in F_t$ be the hypothesis in $F_t$ with the largest index $n \leq t$. Output the distribution $\mu_t$ over $\text{supp}(h_n) \setminus x_{1:t}$ that witnesses $h_n$'s $\alpha$-feasibility.

We now show that the generator $\mathcal{G}$ that follows this mapping and outputs $\mu_t$ at step $t$ for all $t \in \mathbb{N}$ satisfies representative generation in the limit. To do this, we need to verify that $\mathcal{G}$ is $\alpha$-representative, and for any enumeration $x_1, x_2, ...$ of an $h \in \mathcal{H}$ and there exists some $t^* \in \mathbb{N}$ such that $\mathcal{G}$ is consistent after timestep $t^*$.

**Property 1: $\alpha$-Representative.** We first show that the generator's output is representative at every $t \in \mathbb{N}$. This is trivially true if $\mathcal{G}$ outputs $\mu_t = \overline{x_{1:t}}$ at Step 3, and by the definition of $\alpha$-feasibility, the $\mu_t$ output at step 4 also $\alpha$-approximates the empirical distribution of groups. Thus, in either case the $\mu_t$ output satisfies the representation requirement. Because this holds true for any datastream $x_1, x_2, ...$, we conclude that $\mathcal{G}$ is $\alpha$-representative.

**Property 2: Consistent.** It remains to show that there exists some $t^* \in \mathbb{N}$ such that for all $s \geq t^*$, $\text{supp}(\mu_s) \subseteq \text{supp}(h) \setminus x_{1:s}$. Let $t \in \mathbb{N}$ be the timestep guaranteed to exist by Lemma 4.6 such that for all $s \geq t$, $h$ is a critical hypothesis, i.e. $h \in C_s$. Let $d \in \mathbb{N}$ be the finite timestep guaranteed to exist by Lemma 4.8 such that for all $s \geq d$, $h$ is $\alpha$-feasible. Let $t^* = \max\{d, t\}$. It follows that for all $s \geq t^*$, we have $h \in F_s$, as it is both critical and $\alpha$-feasible.

Thus, for any timestep $s \geq t^*$, we are guaranteed that $F_s$ is non-empty, and $\mu_s$ is guaranteed to satisfy $\text{supp}(\mu_s) \subseteq \text{supp}(h_n) \setminus x_{1:s}$, where $h_n \in F_t$ is the hypothesis with the largest index $n \leq t$. It follows from the definition of a critical hypothesis that we must have $\text{supp}(h_n) \subseteq \text{supp}(h)$, and thus $\mu_s$ satisfies consistency.

Thus, we have shown that $\mathcal{G}$ as described above is an $\alpha$-representative generator and is consistent for all $s \geq t^*$, and thus because $\alpha > 0$ was chosen arbitrarily, we conclude that $(\mathcal{H}, \mathcal{A})$ is generatable in the limit with representation. $\square$

### D.4. Proof and Discussion of Lemma 4.9

Before providing the proof of Lemma 4.9, we provide some additional discussion of the result and comparison to the positive result of Kleinberg & Mullainathan (2024).

The algorithm of Kleinberg & Mullainathan (2024) that generates in the limit using only a finite number of membership queries at each step crucially relies on the UUS assumption, in particular the fact that finding an element from $\text{supp}(h) \setminus x_{1:t}$ is always possible because $|\text{supp}(h)| = \infty$, and thus one can simply enumerate the elements of $\mathcal{X}$ until an unseen point is encountered. In the case of representative generation, however, a generator may need to generate from $(\text{supp}(h) \cap A_i) \setminus x_{1:t}$, which is not always guaranteed to be infinite, making this problem a lot more difficult. We note that in the case where $\mathcal{A}$ partitions $\mathcal{X}$, if we made the strong assumption that $\text{supp}(h) \cap A_i$ is either empty or has infinite size for all $A_i \in \mathcal{A}$ and $h \in \mathcal{H}$, then this would be similar to the assumption made by Kleinberg & Mullainathan (2024), and we could use membership queries to generate with representation using an algorithm almost identical to their membership-query algorithm.

However, the Lemma 4.3 shows that weakening this assumption to allow for groups with finite intersection with hypotheses in $\mathcal{H}$ introduces some inherent difficulty. In particular, no algorithm that works even just for very simple, finite pairs of $\mathcal{H}$ and $\mathcal{A}$ can generate in the limit with representation using only membership queries.

With this in mind, we present the proof below.

*Proof of Lemma 4.9.* Consider a deterministically computed randomized generator $\mathcal{G}$ that at step $t$, sees examples $x_1, ..., x_t$ and can make queries of the form "$x \in \text{supp}(h)$" or "$x \in A_1$?". Note that because we only have two groups making up the partition, querying about $A_2$ as well can provide no extra information.

Assume for contradiction that at each timestep $t$, the generator makes a finite number of such membership queries before outputting its generated distribution $\mu_t$, and satisfies representative generation in the limit.

We examine the execution of this generator and use it to construct a hypothesis $h$, partition $\mathcal{A}$, and enumeration $x_1, x_2, ...$ of $\mathsf{supp}(h)$ that forces the generator to make infinitely many mistakes.

Let $u_1, u_2, ....$ be an enumeration of $\mathcal{X}$.

We now imagine running $\mathcal{G}$ (this can be run offline prior to execution as the generator is deterministic) and use the run to build up a hypothesis $h$, an enumeration $k_1, k_2, ...$ of $\mathsf{supp}(h)$, and disjoint groups $A_1, A_2$ with $A_1 \cup A_2 = \mathcal{X}$.

We introduce some variables for keeping track of our constructed example:

- A dictionary $H : \mathcal{X} \to \{-1, 0, 1\}$ that keeps track of the values assigned for the language $h$ thus far, with $H(x) = 1$ if $h(x) = 1$, $H(x) = 0$ if $h(x) = 0$, and $H(x) = -1$ if the value has not been assigned. We initialize $H(x) = -1$ for all $x \in \mathcal{X}$.

- A dictionary $A : \mathcal{X} \to \{-1, 1, 2\}$ that keeps track of the group membership of each $x$ (with 1 and 2 responding to $A_1$ and $A_2$, respectively, and $-1$ if $x$ has not yet been assigned). We initialize $A(x) = -1$ for all $x \in \mathcal{X}$.

- A list $K$ that will keep track of the enumeration of elements. We begin with $K = \{\}$.

- A queue $Q$, to which we can insert elements and pull them off the queue in a first-in, first-out manner.

Now, at each timestep $t = 1, 2, ...$, we perform three stages of actions. The first is to add to the enumeration, the second is to answer the queries of the generator, and the last is to handle the generator's outputted distribution $\mu_t$ for that timestep.

**Stage 1: Adding to the enumeration.** If $t$ is odd or $Q$ is empty, we find the smallest $i \geq 1$ such that $A(u_i) = -1$. Such a $u_i$ is guaranteed to exist as $u_1, ...$ is an enumeration of $\mathcal{X}$, which starts with infinitely many $u_i$ with $A(u_i) = -1$, and at each step we will only fix the membership of a finite number of such $u_i$. Having found this $u_i$, we set $A(u_i) = 1$, $H(u_i) = 1$, and append it to the enumeration $K$, so set $k_t = u_i$.

Otherwise, if $t$ is even and $Q$ is non-empty, we pull an element $x$ off of the queue, and append it to $K$, thus setting $k_t = x$.

**Stage 2: Answering $\mathcal{G}$'s queries.** We now run the generator upon seeing $k_1, ..., k_t$, and answer each membership query as follows.

Every time the generator asks "$x \in \mathsf{supp}(h)$?" for some $x$, if $H(x) \neq -1$, and the membership has already been fixed, we output the correct membership value. Otherwise, if $H(x) = -1$, we answer "yes," set $H(x) = 1$, $A(x) = 2$, and add it to $Q$.

Every time the generator asks "$x \in A_1$?," if $A(x) \neq -1$, we provide the correct answer that has already been fixed. Otherwise, we set $A(x) = 2$, $H(x) = 1$, and add it to $Q$. Note that if $A(x) = -1$, we necessarily have $H(x) = -1$.

**Stage 3: Handling $\mathcal{G}$'s output.** After a finite number of queries, $\mathcal{G}$ is guaranteed by assumption to output a distribution $\mu_t$. We make no assumptions about the representation of $\mu_t$, and thus it could have infinite support.

We perform the following actions depending on the contents of $\mathsf{supp}(\mu_t)$:

- If there exists an $x \in \mathsf{supp}(\mu_t)$ with $H(x) = 0$ or $x \in \{x_1, ..., x_t\}$, we do nothing and end execution for the timestep.

- If there exists an $x \in \mathsf{supp}(\mu_t)$ with $H(x) = -1$, i.e. $x$ has not yet been queried by the generator, we set $H(x) = 0$, $A(x) = 2$, and finish execution for timestep $t$.

- Otherwise, note that every $x \in \mathsf{supp}(\mu_t)$ has $H(x) = 1$, $A(x) = 2$ by definition of our construction.

We repeat these three steps for each timestep. This completes the definition of the construction.

We now verify the necessary facts for this $\mathcal{H}$, enumeration, and group partition to be well-defined. In particular, $\mathcal{H} = \{h\}$ must be $UUS$ and $K$ must be a valid enumeration of $\mathsf{supp}(h)$. Clearly, each $x$ can be assigned only one of $A(x) = 1$ or $A(x) = 2$, so we have a valid partition.

We first consider the UUS property of $\mathcal{H}$. We consider two cases. First, if the generator makes a finite number of queries at each time step as promised, then $\{x \in \mathcal{X} : H(x) = -1\}$ will have infinite size at all timesteps, and so at every timestep we are able to find an unseen $x \in \mathcal{X}$ with $H(x) = -1$ to add to the enumeration, or pull one off the queue. Because this can be repeated indefinitely, $\mathsf{supp}(h)$ will be infinite. On the other hand, if at some finite timestep the generator makes an infinite number of queries, either an infinite number of the queries are on elements with $H(x) = -1$ or $A(x) = -1$, in which case each of these elements get set to $H(x) = 1$ resulting in an infinite support, or there are only a finite number of such queries, and thus the remaining un-queried portion of $\mathcal{X}$ must be infinite, and we can set all of it to $H(x) = 1$ to again get an infinite support. Thus, in all cases $\mathsf{supp}(h)$ satisfies the UUS property.

Lastly, we show that $K$ is a valid enumeration. First, consider the case where at some timestep $t$, the generator makes an infinite number of queries. We can follow the assignment of $H$ above to construct a UUS $\mathcal{H}$, and then append any enumeration of the remaining unseen elements with $H(x) = 1$ to $K$ to obtain a valid enumeration.

Otherwise, we can assume that at every step, the generator makes a finite number of queries. Clearly by definition of the construction, only elements with $H(x) = 1$ are added to the enumeration, meaning that $\bigcup_{i \in \mathbb{N}} \{k_i\} \subseteq \mathsf{supp}(h)$. Thus, it remains to show that the $K$ covers all of $\mathsf{supp}(h)$. Consider any $x \in \mathsf{supp}(h)$, i.e. an $x \in \mathcal{X}$ such that there exists some finite timestep at which $H(x)$ is set to 1. We will show that there exists some $j \in \mathbb{N}$ such that $k_j = x$. Note that because $\mathsf{supp}(h) \subseteq \mathcal{X}$ and $u_1, \ldots$ is an enumeration of $\mathcal{X}$, there exists some $i \in \mathbb{N}$ such that $u_i = x$. At timestep $2i - 1$, we have three possibilities.

- $k_{2i-1} = u_i$.

- $k_{2i-1} \neq u_i$, but there exists a $j < 2i - 1$ such that $k_j = u_i$.

- $k_{2i-1} \neq u_i$, and $u_i$ has not appeared earlier in the enumeration.

If we are in either of the first conditions, we are done because we have guaranteed that there exists a $j \in \mathbb{N}$ such that $k_j = x$. Note that in the third case, this can only happen because $u_i$ is currently in the queue. Because at each previous step the generator made a finite number of membership queries, the position of $u_i$ in the queue must be some finite $s \in \mathbb{N}$. Thus, because even timestep outputs an element from the queue if it is nonempty, we are guaranteed that $k_{2i-1+2s-1} = u_i$, and thus in all cases there exists a $j \in \mathbb{N}$ such that $k_j = x$, and thus $K$ is a valid enumeration of $\mathsf{supp}(h)$.

**Construction of Contradiction.** We finally consider what happens when the generator is run on enumeration $K$ with group and hypothesis membership defined by $A$ and $H$ as above. Consider any timestep $t$, where the generator outputs a distribution $\mu_t$.

We consider the three cases considered in Stage 3. Note that in both of the first two cases, the generator must violate consistency, as there exists an $x \in \mathsf{supp}(\mu_t)$ with $x \notin \mathsf{supp}(h) \setminus \{x_1, \ldots, x_t\}$.

Otherwise, the only other possibility is given by the third case, which guarantees that every $x \in \mathsf{supp}(\mu_t)$ has $A(x) = 2$, and thus $x \in A_2$. This means that the generator is not representative of the data sequence thus far, because by definition of the enumeration, $\overline{x_{1:t}}|_{\mathcal{A}}(1) \geq 1/2$ while $\mu_t|_{\mathcal{A}}(1) = 0$, and thus $\|\overline{x_{1:t}}|_{\mathcal{A}} - \mu_t|_{\mathcal{A}}\|_\infty \geq 1/2 > \alpha$.

Thus, the generator fails to generate consistently with group constraints at every timestep. We thus conclude that the assumption was false, and at some iteration the generator must make an infinite number of membership queries. $\square$

