# OpenReview forum: "Representative Language Generation"
_ICML.cc/2025/Conference — ICML 2025 poster_

### Official Review · Reviewer_y5Xy · 2025-03-13

**Overall Recommendation:** 3

**Summary:**

This paper introduces a theoretical framework to characterize generative models’ capacities/abilities to produce samples that reflect the diversity seen in the data whose distribution the model is trying to approximate. Such characterizations are of interest to the machine learning community as they let us quantitatively define whether different groups in the support of the data-generating distribution are properly represented by the model of interest. The paper outlines the criteria by which a generator fulfills different degrees of representative-ness (these build off of definitions given in prior work) and then consider the theoretical feasibility of different models to meet these criteria. They provide information-theoretic bounds and computational bounds (where possible). They present both positive and negative results for whether representative generation can be fulfilled for different generator–hypothesis class pairs (e.g, that for a certain hypothesis class + alpha, no generator can achieve representative generation in the limit using only a finite number of membership queries).

**Claims And Evidence:**

Yes, main claims are stated as theorems, corollaries, and lemmas, which are supported by proofs (outlines of proof are given in body of paper and formal proofs in appendix)

**Essential References Not Discussed:**

There are several works on generative model evaluation that use notions of precision and recall (e.g, Sajjadi et. al. 2018, Kynkäänniemi et. al. 2018) in the attempt to assess whether models capture the diversity of the data-generating distribution. These should probably be discussed

**Experimental Designs Or Analyses:**

N/A

**Methods And Evaluation Criteria:**

There aren’t really any evaluations in this paper, as its providing a theoretical framework.

**Other Comments Or Suggestions:**

Consider using fewer scare quotes. These are typically used to indicate the inaccurate use of a term. Given that the merits of this paper are a theoretical framework, if you need a different term than the one you’re putting scare quotes around, I would recommend defining such a term.

**Other Strengths And Weaknesses:**

The paper doesn’t offer any insights/recommendations that let the reader bridge the gap between its theoretical results and various practical implementations. To me, this is a major shortcoming. It’s unclear how much the specificity of the assumptions (e.g., UUS and finite support) behind the results limit their direct applicability to real-world systems.

**Questions For Authors:**

Following from the last comment, why is “sample complexity” in quotes throughout the paper? Is the notion not actual sample complexity? This wasn’t evident to me

**Relation To Broader Scientific Literature:**

The paper expands on other works’ attempts to formalize diversity and representativeness in generative models. They discuss the relationship to concepts such as mode collapse ad notions from algorithmic fairness, such as multiaccuracy and multicalibration. I am unfamiliar with the prior related work but the claimed extension this work offers seems worthwhile

**Theoretical Claims:**

There are numerous theoretical claims in the paper. I was unable to check all rigorously. The informal arguments given in the paper (formal proofs were in appendix) intuitively made sense

---

> ### Author Rebuttal · Authors · 2025-03-30
>
> We thank the reviewer for their helpful comments. We address the reviewer's concerns below.
>
> > There are several works on generative model evaluation that use notions of precision and recall (e.g, Sajjadi et. al. 2018, Kynkäänniemi et. al. 2018) in the attempt to assess whether models capture the diversity of the data-generating distribution. These should probably be discussed
>
> We thank the reviewer for pointing out these relevant works. We will make sure to cite them in the camera-ready version.
>
> > The paper doesn’t offer any insights/recommendations that let the reader bridge the gap...
>
> We emphasize that our paper primarily offers theoretical contributions by extending and analyzing the model introduced by Kleinberg and Mullainathan. For insights regarding practical applications and recommendations, we direct you to our response to Reviewer U9Ez above (Re: Practical applicability of results) addressing these aspects.
>
> Regarding the specific assumptions in our framework, the UUS assumption is standard across generation literature and serves a fundamental purpose: ensuring generators can indefinitely produce novel elements from the true language. This assumption aligns with practical contexts, where the set of "valid generations" is effectively infinite--for instance, the set of all valid English passages is unbounded. The finite support property is similarly incorporated to ensure basic feasibility. As demonstrated in Lemma 4.3, without this assumption, representative generation becomes impossible even when the generator has complete knowledge of the true language.
>
> > Consider using fewer scare quotes
>
> Thanks for pointing this out. We will make sure to dial back on the use of scare quotes in the final version.
>
> > Following from the last comment, why is “sample complexity” in quotes throughout the paper?
>
> Yes, the notion for uniform and non-uniform generation, is indeed the actual sample complexity. We will remove the scare quotes in these sections.

---

### Official Review · Reviewer_U9Ez · 2025-03-16

**Overall Recommendation:** 4

**Summary:**

The paper defines a new property of generators called “representative generatability” to provide a theoretical framework for comparing a generator's representation (occurrence) of distinct groups present in the training distribution with the representation in the distribution of output sequences. The contribution is well-motivated with real-world bearing on issues observed in practice in generative systems (like LLMs), such as bias propagation and mode collapse. Results are presented in three settings derived from prior works –– uniform generation, non-uniform generation, and generation in the limit.

**Claims And Evidence:**

Yes

**Essential References Not Discussed:**

I am not aware of any missing related works that have not been cited.

**Experimental Designs Or Analyses:**

Not applicable since the work is a theoretical contribution.

**Methods And Evaluation Criteria:**

Not applicable since the work is a theoretical contribution.

**Other Comments Or Suggestions:**

None.

**Other Strengths And Weaknesses:**

**Strengths:**

The paper focuses on the problem of diverse generation, a topic of high-relevance in practice, and presents a theoretical framework to analyze the problem. Given the abundance of iterative methodological contributions in the area, I think this is a useful contribution for the community to assess the theoretical limits of what we want from such systems.

**Weaknesses:**

While I understand that the work is a purely theoretical contribution, given that the motivation uses practical systems (such as LLMs) to establish impact, it would have been useful to present a discussion about what the authors think might be the practical (and immediate, if any) implications of their results.

**Questions For Authors:**

L205-208: Could you discuss the choice of the supremum distance as the measure of "closeness" versus other distances? The choice currently seems fairly unjustified.

**Relation To Broader Scientific Literature:**

This work directly follows from [1], [2], [3], [4], and [5], and provides a useful new lens that has real-world relevance to current generative systems in practice (albeit with no obvious takeaways for practitioners).

[1] Kleinberg, J. and Mullainathan, S. Language generation in the limit.
[2] Li, J., Raman, V., and Tewari, A. Generation through the lens of learning theory.
[3] Kalavasis, A., Mehrotra, A., and Velegkas, G. Characterizations of language generation with breadth.
[4] Charikar, M. and Pabbaraju, C. Exploring facets of language generation in the limit.
[5] Kalavasis, A., Mehrotra, A., and Velegkas, G. On the limits of language generation: Trade-offs between hallucination and mode collapse.

**Theoretical Claims:**

My reading of the proofs revealed no errors.

---

> ### Author Rebuttal · Authors · 2025-03-30
>
> > Re: Practical applicability of results
>
> Thank you for your question. While we maintain that this is primarily a theoretical work, we've addressed how our research may or may not relate to practical applications in our global comment to all reviewers above. We will plan to incorporate more of this discussion in the final version of the paper.
>
> To clarify our position, our work extends the theoretical model introduced by Kleinberg and Mullainathan, which—along with subsequent research discussed in our introduction—examines generation in a worst-case scenario without making assumptions about data distribution or learning methods. This approach deliberately highlights fundamental tensions and possibilities in generative tasks from a theoretical perspective rather than focusing on immediate practical implementations.
>
> That being said, our work on representative generation can be viewed as an extension of Kleinberg and Mullainathan's model of generation, aiming to align it more closely with real-world generation approaches and values. Specifically, while real-world approaches typically aim to develop generative models that closely approximate training distributions (as reviewer 2 noted)—and it is indeed natural to expect our generations to resemble training data in certain aspects—Kleinberg and Mullainathan's notion of generation in the limit imposes no requirement that generated data must resemble previously observed data, only that it must belong to the true language. Our work maintains the generation-in-the-limit framework while introducing an additional constraint: generations must resemble training data with respect to simple statistical tests measuring the prevalence of certain subpopulations. Arguably, our notion of representation is useful to formalize even in a practical setting, as it addresses an important consideration: generative models can potentially under- or over-represent certain subpopulations, even when they demonstrate good overall alignment with the training data.
>
> At a high-level, we view the main contributions/takeaways of our work as two-fold, in the context of existing works on generation:
> 1) Our additional constraint of representational generation highlights key tensions and possibilities between the positive results of Kleinberg and Mullainathan's model and real-world approaches to generation. In particular, we show that there are many settings where generation in their model is possible, but becomes impossible when the generative model is additionally required to satisfy representation. On the other side, some of our positive results signal that matching training data makeup is not always in conflict with the goal of generation. For instance, we show that requiring representation with respect to any finite set of groups is no more difficult than just generating in the limit for any countable class $\mathcal{H}$.
> 2) Our work is among a recent line of work attempting to understand the trade-offs between obtaining novelty and breadth in language generation. Like these works, some of our results are negative: if one cares about computability, then achieving novelty and representativeness is impossible. However, from a sample-efficiency perspective, in contrast to the many negative results on generating with breadth, our formulation of representation offers a tractable relaxation of breadth that maintains semantic relevance while remaining feasible across many natural classes of languages and groups.
>
> > Re: Choice of supremum distance
>
> Our selection of the supremum distance draws directly on the foundational principles established in algorithmic fairness literature, which emphasizes the importance of limiting the error experienced by the worst-off group. This perspective prioritizes ensuring good representation for every group rather than merely optimizing for average performance. The supremum distance naturally operationalizes this principle by measuring the maximum disparity across all groups, effectively placing an upper bound on the error that any group might experience.
>
> It's worth noting that for finite group settings, different choices of distance measures (such as $L_1$, $L_2$, or the supremum distance (L$_\infty$)) are typically within constant factors of one another, making the specific choice less critical. However, as we move to settings with infinite groups, these equivalences break down—distance measures such as $L_1$ become less informative, potentially obscuring significant disparities among individual groups.
>
> As we highlight in our concluding discussion, while we selected the supremum distance for its robust guarantees from an algorithmic fairness perspective, exploring representation guarantees under alternative distance metrics is certainly an interesting direction for future research. In our revised paper, we will provide a more thorough justification for our choice of the supremum distance as an appropriate measure of closeness within our framework.

---

> > ### Comment · Reviewer_U9Ez · 2025-04-05
> >
> > Thank you for the helpful discussion. I'd be happy to increase my score and hope the authors include aspects of their discussion in their final paper.

---

### Official Review · Reviewer_JWgs · 2025-03-17

**Overall Recommendation:** 3

**Summary:**

This work introduces the concept of "Representative Generation" and its variants, aiming to provide a theoretical framework that characterizes generators (i.e. generative models) such that, when the training data distribution consists of multiple groups of interest, the generator outputs closely approximate the proportions of data across the groups. In contrast, prior work proposes the "generation in the limit" characterization of generators, where generators can get away by generating from a restricted subset of the ground-truth. While prior work shows that generation in the limit, which is much weaker than representative generation, can be achieved in many scenarios, this paper shows that representative generation is impossible using only a finite number of membership queries, i.e. querying whether  an element belongs to a certain group.

**Claims And Evidence:**

Yes, to the best of my judgement.

**Essential References Not Discussed:**

N/A

**Experimental Designs Or Analyses:**

N/A

**Methods And Evaluation Criteria:**

N/A

**Other Comments Or Suggestions:**

N/A

**Other Strengths And Weaknesses:**

N/A

**Questions For Authors:**

1. One common paradigm for training generative models is to minimize KL(ground-truth | generative model), and in this case, it is impossible for a generative model to get away with generating only cat images when the training data consists of 1/3 cats, 1/3 dogs and 1/3 rabbits. Could you use this example to help clarify, for a non-expert like me, how is "representative generation" different from "generation in the limit" and how to interpret the negative result (Lemma 4.9) proved in this work?
2. Could you help clarify what "at each step" is referring to in `Rather than generating a single element at each step, a representative generator generates a distribution over multiple elements at each step.`

**Relation To Broader Scientific Literature:**

As suggested by the author, the theoretical analysis is relevant to the commonly observed/discussed phenomenon of generative models often exacerbating biases presented in training data.

**Theoretical Claims:**

I do not have the required expertise to verify the correctness of the proofs.

---

> ### Author Rebuttal · Authors · 2025-03-30
>
> We thank the reviewer for their comments.
>
> > Could you use this example to help clarify, for a non-expert like me, how is "representative generation" different from "generation in the limit" and how to interpret the negative result (Lemma 4.9) proved in this work?
>
> In the original notion of "generation in the limit" proposed by Kleinberg \& Mullainaithan (2024) the goal of the generator was to only eventually produce new, valid examples. In our model of "representative generation", the goal of the generator is to not only produce new, valid examples, but to produce them in a way that is "representative" of the input training data. So now, the goal is not only to produce new, valid examples, but to produce new, valid examples that align with the properties/preferences of the training data. The negative result (Lemma 4.9) shows that if one cares about computability, then representative generation is strictly harder than generatability in the limit -- there are cases where once can produce new, valid examples, but not in a way that aligns with the properties/preferences of the training data.
>
> You raise a good point that the objective of generation in the limit is somewhat different from the standard objective in practice of minimizing distance to training data. Please see our post to Reviewer U9Ez below (Re: Practical applicability of results) addressing this discrepancy between the theoretical model and practice.
>
> > Could you help clarify what "at each step" is referring to in Rather than generating a single element at each step, a representative generator generates a distribution over multiple elements at each step.
>
> In this paper (as well previous papers in this line of work), we consider a game between a generator $G$ and an adversary $A$. This game is played sequentially over rounds $t=1, 2, \cdots.$ In each round $t \in \mathbb{N}$, the adversary reveals to the generator a new example $x_t \in X$. Upon observing $x_t$, the Generator must output a new example $\hat{x}_t.$ In previous works, the generator $G$ was deterministic. In our work, $G$ is randomized and thus produces a distribution $\mu_t \in \Delta X$ after observing $x_t$. So, the "at each step" is referring to the rounds $t = 1, 2, ...$ .

---

### Official Review · Reviewer_g3vz · 2025-03-17

**Overall Recommendation:** 3

**Summary:**

This paper extends the study of the recent model of language generation introduced by Kleinberg and Mullainathan in a 2024 paper. In this model, this paper defines a notion of representative generation where the generator is roughly speaking required to assign similar probability masses to each group A (from a class $\\mathcal{A}$) as is assigned by the sequence of examples provided by the adversary. The paper provides several results exploring when different notions of generation (as defined by Li et al. (2024)) can be achieved with the additional representation constraint.  Their first two sets of results consider the case where groups in $\\mathcal{A}$ are disjoint.

1. Their first result is a characterization of representative uniform generation under the disjointness assumption. This characterization builds on a characterization of Li et al. (2024). Next, we list some examples and results complementing this characterization:
   1. This characterization, in particular, shows that representative uniform generation is always possible with a finite language collection and finitely many groups.
   2. Complementing this, they also show that representative uniform generation is strictly harder than uniform generation (by providing a class that is uniform generatable but not representative uniform generatable with even two groups).
   3. Further, they also give another example where $\\mathcal{A}$ is countably infinite and where a hypothesis class of size 1 cannot be generatable in the limit with representation (a weaker requirement than representative uniform generation).
2. Their second result characterizes representative non-uniform generation under the disjointness assumption. This characterization is identical to the characterization of non-uniform generation by Li et al. (2024) except that one needs to substitute the characterization of uniform generation with that for representative uniform generation.
   1. This characterization, in particular, shows that all countably infinite classes with a finite partition  $\\mathcal{A}$  are representatively non-uniformly generatable.
3. Their final set of results for generation in the limit drops the disjointness assumption, but makes a different “finite support” assumption in Definition 4.2.
   1. Their main result for generation in the limit is that any countably infinite class with a countably infinite $\\mathcal{A}$   satisfying Definition 4.2 is generatable in the limit with representation.
   2. Further, they also demonstrate computational barriers to generation with only membership queries by showing that no algorithm can generate in the limit with representation using only finitely many membership queries even when the hypothesis class has a single hypothesis and there are only two two groups.

**Post-rebuttal Update** Dear authors, thank you for your response and for explaining that the group closure dimension has the finite character property. The rebuttal addresses most of my concerns, I maintain my original rating of weak accept. I do not give a stronger accept as I am not 100% sure about the gap in technical novelty compare to prior work of Li et al.

**Claims And Evidence:**

Yes, the claims in the paper are supported by the rigorous proofs.

**Essential References Not Discussed:**

To the best of my knowledge, the paper discusses relevant prior works.

**Experimental Designs Or Analyses:**

Not applicable.

**Methods And Evaluation Criteria:**

Not applicable.

**Other Comments Or Suggestions:**

I found the Finite Support Size assumption (Definition 4.1) hard to parse. As a sanity check, it seems to make sense in the case where there are finitely many groups. I think it would be very useful to include several (simple) examples where the Finite Support Size assumption holds and when it is violated.

Typos:

1. Line 90 “Representative Uniform Generation,” \-\> “Representative Uniform Generation.”

**Other Strengths And Weaknesses:**

The paper studies an interesting and fundamental learning theoretic model for generation. The problem they study seems like a natural extension of existing works.

**Questions For Authors:**

I have some questions for the authors. The most important ones are Q2 and Q3.

First, for the closer dimension introduced by Li et al. (2024), it is relatively easy to certify that a hypothesis class satisfies the definition. In some sense, the definition was “interpretable.” The definition of the group closure dimension (which characterizes representative uniform generation under the disjointness assumption) seems less interpretable, at least as currently stated.  (Q1) Is there an easy way to check if a hypothesis satisfies group closure dimension? How can one interpret it?

I understand that the group closure dimension characterizes the sample complexity of the task, but it was not easy to parse what it means. If there is a better way to state it, that would be very useful for the readers.

My second question is: (Q2) Could the authors shed some light on why it is challenging to characterize or provide sufficiency conditions for uniform and non-uniform generation without the disjointness assumption on $\\mathcal{A}$?

Finally, for the result on computationally barriers with only membership queries. (Q3) How are the techniques used in this result different from the techniques of Charikar and Pabbaraju? Are the techniques the same? If not, can I find a discussion about the differences somewhere in the paper?

**Relation To Broader Scientific Literature:**

The paper advances a line of work studying language generation in a model recently introduced by Kleinberg and Mullainathan in a 2024 NeurIPS paper. The present paper introduces the task of representative language generation which has not been considered in other works studying this model of language generation.

**Theoretical Claims:**

I skimmed the proofs but did not check their correctness.

---

> ### Author Rebuttal · Authors · 2025-03-30
>
> We thank the reviewer for finding that our paper studies an interesting and fundamental learning theoretic model for generation. We address the reviewer's comments/questions below.
>
> > Finite Support Size Assumption
>
> We can certainly add some examples to clarify the definition. We'll highlight a few simple examples here that we can add to the final version to add intuition for the assumption:
> - As you correctly observe, any collection of finite groups will always satisfy the finite support size assumption.
> - When the collection of groups is infinite and disjoint, the assumption simplifies to the requirement that for any $h \in \\mathcal{H}$, only a finite number of groups have a finite size intersection with $\mathsf{supp}(h)$.
> - A simple example of a collection of groups that does not satisfy the assumption is any infinite collection of groups where the size of every group is finite, such as the collection of all singletons $\mathcal{A} = \lbrace{\lbrace{x\rbrace} : x \in \mathcal{X}\rbrace}$, or the collection defined in the proof of Lemma 4.3.
>
> > (Q1) Is there an easy way to check if a hypothesis satisfies group closure dimension? How can one interpret it?"
>
> Like other combinatorial dimensions (e.g VC and Littlestone dimension), it is likely not possible to efficiently compute the group closure dimension at a fixed scale. That said, the group closure dimension does satisfy what is known as the Finite Character Property [1]:  for every $d \in \mathbb{N}$ and class $\mathcal{H}$, the statement $\operatorname{GC}_{\alpha}(\mathcal{H}) \geq d$ can be demonstrated by a finite set of domain points in $\mathcal{X}$ and a finite collection of members of $\mathcal{H}.$ This property is also satisfied by most other combinatorial dimension in learning theory literature. In terms of interpretation, intuitively, one should think of the group closure dimension as measuring the maximum number of samples one needs to see until they are guaranteed a winning strategy. Here, a winning strategy is one where there exists a distribution over examples which is consistent and representative. It turns out that the group closure dimension quantifies exactly this in a way that does not explicitly require quantifiers over distributions of the example space by exploiting the disjoint nature of the groups.
>
> [1] Ben-David, Shai, et al. "Learnability can be undecidable." Nature Machine Intelligence (2019)
>
> > (Q2) Could the authors shed some light on why it is challenging to characterize or provide sufficiency conditions for uniform and non-uniform generation without the disjointness assumption on ?
>
> There are several issues that arise when one tries to go beyond disjoint groups.
> - For one, the distribution-free definition of the group closure dimension heavily relies on the fact that the groups are disjoint. Thus, while it is possible to define a version of the group closure dimension for arbitrary groups, it is likely that it will be abstract and not satisfy the Finite Character Property [1].
> - The second issue is that when the groups are not a partition of the domain $\mathcal{X}$, the vector of induced probabilities (Definition 2.5) is no longer guaranteed to be a probability distribution. For example, consider a sequence of examples $x_1, x_2, ..., x_d$ contained in every group. Any dimension that gives a characterization will likely have to have quantifiers iterating over entire group vectors instead of individual groups. This makes the dimension hard to parse and less meaningful. In addition, with overlapping groups, the dimension will need to take into account arbitrary intersections of these groups, which again significantly increases the complexity.
>
> > (Q3) How are the techniques used in this result different from the techniques of Charikar and Pabbaraju?
>
> We briefly compare the two proofs in the beginning of Section 4.2, but will provide a more detailed discussion here that we will include in the final version of the paper.
>
> Both proofs employ the same fundamental approach to construct adversarial counterexamples and prove impossibility: they develop strategies that, given a generator, methodically build an adversarial enumeration that forces the generator to violate its promised guarantee. Charikar and Pabbaraju focus on the impossibility of non-uniform generation guarantees, while our work examines representative generation in the limit. Importantly, without the representation requirement, generation in the limit is always achievable with membership queries, as demonstrated by Kleinberg and Mullainathan [2]. Our adversarial construction specifically exploits the additional representation constraint by carefully designing groups that force the generator to violate either representation or consistency. Furthermore, our setting differs in that we consider generators that output distributions rather than single elements.
>
> [2] Kleinberg, Jon, and Sendhil Mullainathan. "Language generation in the limit." (2024)

---

### Decision · Program_Chairs · 2025-05-01

**Decision:**

Accept (poster)

**Comment:**

The paper studies representative language generation, extending recent theoretical frameworks. It characterizes conditions under which representative generation can or cannot be achieved, providing insightful theoretical bounds and impossibility results. Reviewers generally agree that the paper addresses an interesting and meaningful extension of prior work and contributes rigorous theoretical analyses. Given the overall positive reviewer consensus, I recommend acceptance for this paper.